# Hemodynamics in chronic pain: A pathway to multi-modal health risks

**Dmitry M. Davydov**[1]*, **Carmen M. Galvez-Sánchez**[2], **Gustavo A. Reyes del Paso**[3]

**1** María Zambrano senior scholar, University of Jaén, Jaén, Spain, **2** Department of Personality, Evaluation and Psychological Treatment, University of Murcia, Murcia, Spain, **3** Department of Psychology, University of Jaén, Jaén, Spain

* d.m.davydov@gmail.com

**Data Availability Statement:** Data availability statement: Data supporting this study are not currently publicly available due to its farther use by the research team. Please contact the corresponding author (DMD). Data requests may

## Abstract

Pain-o-metry provides biomarkers indicating connections between pain-related conditions and the health of various physiological systems, including the cardiovascular system. In this study, a non-linear data-driven analytical technique was employed to analyze second-by-second hemodynamic responses to recurrent clino-orthostatic challenges in 43 female fibromyalgia patients and 38 healthy women. The primary goal was to elucidate the systemic and systematic interaction of diverse hemodynamic and cardiovascular mechanisms across various timeframes and phases, precisely identifying their collective role as a unified bodily mechanism in managing regular gravity-induced blood redistribution challenges within the context of chronic pain. Findings suggest a connection between chronic disease and the equilibrium between cardiac preload and afterload in blood pressure regulation. Patients exhibit a shift towards afterload due to deficiencies in mechanisms governing initial transient reactions and later steady-state processes related to acutely induced blood redistributions. The imbalance is linked to reduced venous blood return, increasing cardiac strain—particularly in terms of contractility and heart rate—as the body compensates for heightened cardiac afterload and reduced effective blood volume. This makes individuals more susceptible to chronic peripheral tissue and cerebrovascular hypoperfusion, potentially leading to chronic ischemia and inflammation in various tissues and organs. The data-driven analytical technique enables the identification of combinations of gravity-induced hemodynamic and cardiovascular responses within specific timeframes for precise detection. This approach aims to facilitate potential diagnostic and monitoring applications in wearable devices, enhancing the ability to identify at-risk populations for preventive interventions.

## Introduction

Ignoring the close relationships between pain and blood pressure (BP) in treatment guidelines and clinical practice may have a highly negative impact on the present-day control of both pain and BP. These conditions are the most relevant risk factors for disability in the world and the biggest contributors to the global burden of disease and mortality [1–5]. In some patients with hypertension, elevated BP has been found to have analgesic effects as an 'adaptation to

also be directed to the General Coordinator and Secretary of the Ethics Committee, referencing #: OCT.18/1.PRY and the title "Central Sensitization in Chronic Pain: The Case of Fibromyalgia": https://www.ujaen.es/gobierno/vicinv/comision-de-etica.

**Funding:** Gustavo A. Reyes del Paso the Spanish Ministry of Science and Innovation, co-financed by European Regional Development Fund [PID2022-139731OB-I00] The funders had no role in study design, data collection and analysis, decision to publish, or preparation of the manuscript.

**Competing interests:** The authors have declared that no competing interests exist

pain' control mechanism (BP-related hypoalgesia). As a result, the use of anti-hypertensive medications and related lifestyle recommendations may inadvertently worsen pain management in these individuals [2, 5]. However, in other patients with chronic pain, hyperalgesia has been related to BP elevation, and thus effective pain control might have antihypertensive effect for them [2, 5]. The inconsistency in the relationship between pain and BP may be explained by two contrasting hemodynamic mechanisms leading to hypertension: hypovolemia and hypervolemia [6]. These mechanisms respond differently to various antihypertensive medications and lifestyle recommendations, with varying potential effects on pain sensitivity and the pathways to chronic pain, aspects that prior studies in this field have often overlooked [7]. There is strong experimental and clinical evidence linking nociception and BP regulation, along with proposed central and peripheral interconnections between acute pain regulation, chronic pain development, and cardiovascular (CV) regulation, as discussed in detail elsewhere [1, 5, 7–10]; however, further investigation into the underlying mechanisms of these interactions is needed before this evidence can be effectively translated into personalized clinical practice.

Rapid (within seconds) and delayed (within minutes) hemodynamic responses to challenges have evolved in mammals to maintain BP and effective blood volume within normal ranges, ensuring adequate blood flow and supply to vital organs essential for optimal functioning and survival [11]. These responses include primary mechanisms like central cardiopulmonary and carotid-aortic stretch-receptor reflexes, as well as alternative mechanisms (central, regional, and local autoregulatory reflexes and agents in carotid, cerebral, and renal vessels, and the heart and skeletal muscles) that substitute for primary ones in case of malfunction [12–14]. In emergencies or pathological conditions, these primary reflexes can be augmented, reduced, overridden, or reversed by elements of the CV system activated by alternative mechanisms [12, 15]. For instance, patients with chronic pain exhibit an imbalance in cardiac baroreflex sensitivity, triggering an alternative mechanism of BP regulation via systemic vasoconstriction [8, 9]. This alternative mechanism can result in emergency situations such as recurrent cerebral hypoperfusion during orthostasis, which resemble acute central dehydration events with reduced venous return and stroke volume drop, present as impaired consciousness and falls, and are known as orthostatic intolerance [8, 10, 11, 16]. Orthostatic intolerance, whether hypo-, normo-, or hyper-tensive (i.e., from under- to over-compensated CV responses to these gravity-induced central hypovolemic challenges), is implicated in brain lesions, neuroinflammation, and sustained arterial hypertension later in life and is frequently linked to chronic pain [1, 8, 11, 17–20].

When primary CV reflexes, which regulate BP mainly through blood volume, are impaired, alternative mechanisms that rely on vascular tone for BP control can lead to increased cardiac strain. This often results in negative arousal, as observed in chronic pain, due to abnormal hemodynamic responses to everyday challenges [21–23]. Previous studies have linked the severity of chronic pain to failures in these primary reflexive control mechanisms during such challenges [10, 24]. Nevertheless, the exact temporal coordination of acute hydraulic, hemodynamic, and reflexive CV responses, functioning as components of a unified anti-gravity bodily mechanism, remains unexplored concerning chronic pain pathophysiology. To address this gap, an exhaustive investigation of gravity-induced hemodynamic shifts demands a detailed second-by-second analysis of their swift and gradual, rhythmic and sporadic fluctuations [1, 11, 12].

Historically, various hemodynamic and CV reactions have often been studied as isolated processes, treated separately, and analyzed by averaging across different arbitrary intervals based on proposed specific time and phases in responses [7, 10, 25]. These practices primarily stemmed from limitations of linear statistical methods. For example, linear models were used

to analyze effects, focusing on either minute-by-minute CV activity during the entire clino-orthostatic procedure or second-by-second CV reactivity during specific, hypothesis-driven transition periods between postures [7, 10, 25]. However, these linear models may not fully capture the complexity of physiological responses within and between the hemodynamic and CV systems. These systems are regulated by multiple interacting mechanisms that respond to challenges at different rates—each with its own initiation, duration, recovery, and stabilization times. Furthermore, there are variations both between and within subjects, including delays or advances in these mechanisms, as well as fluctuations during the evoking, adaptive, and recovery phases. These processes are difficult to predict across different populations and in response to various challenges, making it hard for linear statistical methods to capture all of them simultaneously in studies without biasing toward one system, mechanism, or process at the expense of others [7, 10, 25].

The present study aimed to address these limitations by incorporating a non-linear approach alongside conventional linear analysis. Non-parametric modeling techniques, such as smoothing approximation methods using various spline functions, have previously been employed for continuous data analysis and represent results as sets of curves rather than vectors [26]. An automatic, data-driven method was also introduced to adaptively identify the periods and shapes of significant responses, with the added ability to ignore noise and occasional outliers [27]. In this study, one such non-linear approach was designed to avoid conducting multiple linear analyses of predefined short- or long-term periods during clino-orthostatic challenges, which typically rely on fixed temporal resolutions for hemodynamic and CV responses. As a fully automated, data-driven technique, it was expected to capture a broad range of hemodynamic and CV responses—whether passive (e.g., hydraulic), reflexive, or active (cardiac and vascular)—that vary in speed, duration, periodicity, and trend. These responses may include fluctuations and delays that are particularly sensitive to conditions such as chronic pain. The application of advanced non-linear, data-driven analytical techniques in this study was also considered crucial for elucidating the collaborative alterations in diverse hemodynamic processes. These methods can reveal how these processes interact across different time points and phases, displaying varying magnitudes and patterns in response to repeated postural changes between clino- and ortho-static positions.

Thus, this study aimed to clarify how various hemodynamic and CV mechanisms interact as a unified bodily response to physical challenges in the context of chronic pain, highlighting their value as CV 'pain-o-metric' indicators. It also considered potential CV health risks associated with the prolonged extension of these systemic and systematic responses as integrated processes linked to chronic pain. The analysis was conducted impartially and comprehensively in patients with fibromyalgia (FM), a heterogeneous population experiencing chronic pain across various behavioral, somatic, and mental domains. This was contrasted with a healthy control group consisting of pain-free participants to validate the CV 'pain-o-meter' hypothesis (i.e., the potential of identifying chronic pain based on alterations in CV measures and their dynamics in FM patients relative to pain-free individuals) [8–10, 24, 28].

## Materials and methods

### Participants

Forty-three Caucasian women diagnosed with fibromyalgia (FM) by a rheumatologist, meeting the 1990 American College of Rheumatology criteria [29], were recruited between January 20 and May 30, 2019, via the Fibromyalgia Association of Jaén, a local non-commercial organization of FM patients with the aim to improve communication, access to information and healthcare for FM patients living in the Jaén province (southeast of Spain). Exclusion criteria

were the presence of inflammatory, metabolic, cardiovascular, or neurological disorders, severe diseases (e.g., cancer, schizophrenia, or drug abuse disorders), and the use of medications affecting the cardiovascular, peripheral or central nervous system (with the exception of FM-related medications, as detailed below). Daily pain fluctuations and related symptoms, such as sleep quality, were not factored into the scheduling of participants' appointments. The only restriction was the exclusion of patients experiencing an active pain crisis; in such cases, the appointment was rescheduled until after the crisis had subsided. The control (CT) group, recruited during the same period, included 38 healthy women of similar age, nationality, civil status, and educational levels who did not fulfill the same exclusion criteria and had no acute or chronic pain disorders. Controls were not individually matched to patients, and a strict case-control matching procedure was not implemented. However, controls were selected based on the overall sociodemographic characteristics of the patient sample (i.e., within similar ranges of age, weight, and educational levels as reported by the patient group), as patients and controls were evaluated sequentially. Table 1 shows the demographic, symptoms severity,

**Table 1. Demographic parameters, medication use, pain severity and affect estimation scores in fibromyalgia (FM) patients and healthy women (mean [SD] or n [%]).**

| | Fibromyalgia patients (n = 43) | Healthy women (n = 38) | $t$ or $\chi^{2\ a}$ | $p$ |
|---|---|---|---|---|
| Age | 52.40 (8.80) | 49.68 (6.63) | 1.58 | 0.120 |
| Height (m) | 1.60 (0.06) | 1.63 (0.07) | -1.91 | 0.060 |
| Body weight (kg) | 71.84 (13.57) | 68.11 (9.41) | 1.45 | 0.151 |
| Body mass index (kg/m$^2$) | 28.12 (5.21) | 25.74 (3.16) | 2.51 | 0.014 |
| Years of formal education | 11.70 (4.20) | 12.46 (4.11) | -0.82 | 0.416 |
| Civil status (in couple) (n) | 33 (76.7) | 33 (86.8) | 1.36 | 0.243 |
| Smoking (Yes) (n) | 16 (37.2) | 15 (39.5) | 0.04 | 0.834 |
| Daily alcohol consumption (Yes [b]) (n) | 2 (4.7) | 0 (0) | 1.81 | 0.178 |
| Physical activity (Yes [c]) (n) | 25 (58.1) | 12 (31.6) | 5.74 | 0.017 |
| Systolic blood pressure (mmHg) | 105.9 (16.0) | 105.6 (8.7) | 0.08 | 0.933 |
| Diastolic blood pressure (mmHg) | 70.2 (11.4) | 70.0 (7.4) | 0.08 | 0.934 |
| Heart rate (beats per min) | 68.8 (9.0) | 66.6 (8.4) | 1.10 | 0.273 |
| Antidepressant use (n) | 39 (90.7) | 3 (7.9) | 55.40 | < 0.001 |
| Anxiolytic use (n) | 37 (86.0) | 4 (10.5) | 46.03 | < 0.001 |
| Analgesic use (n) | 37 (86.0) | 3 (7.9) | 49.29 | < 0.001 |
| Habitual pain intensity (VAS) | 7.56 (1.71) | 3.00 (1.51) | 12.67 | < 0.001 |
| Current pain intensity (MPQ) | 3.51 (1.08) | 1.11 (0.46) | 13.30 | < 0.001 |
| Total pain (MPQ) | 82.35 (32.24) | 5.38 (8.66) | 15.04 | < 0.001 |
| Depression (BDI-II) | 39.51 (15.44) | 5.24 (6.69) | 13.19 | < 0.001 |
| Trait anxiety (STAI) | 49.93 (16.11) | 24.30 (11.90) | 8.10 | < 0.001 |
| Positive affect (PANAS) | 26.88 (8.88) | 38.46 (8.40) | -5.83 | < 0.001 |
| Negative affect (PANAS) | 33.44 (10.36) | 18.57 (6.98) | 7.42 | < 0.001 |
| Fatigue (FSS) | 51.88 (9.25) | 19.51 (12.54) | 12.96 | < 0.001 |
| Insomnia (OQS) | 33.56 (12.20) | 12.24 (7.72) | 9.46 | < 0.001 |
| Fibromyalgia impact (FIQ) | 70.26 (14.36) | 9.01 (13.03) | 19.99 | < 0.001 |

VAS = visual analogue scale; MPQ = McGill Pain Questionnaire; BDI-II = Beck Depression Inventory-II; STAI = State-Trait Anxiety Inventory; PANAS = Positive and Negative Affect Schedule; FSS = Fatigue Severity Scale; OQS = Oviedo Quality of Sleep Scale; FIQ = Fibromyalgia Impact Questionnaire.

[a] $t$ refers to a two-tailed Student's $t$-test used for analyzing group differences in numerically scaled dependent variables, while $\chi^2$ refers to the non-parametric chi-squared test used for assessing relationships between groups and nominal or ordinal variables.

[b] the identified maximum was one glass of wine per day with meals.

[c] including swimming pool exercises or walking as part of wellness programs for some FM patients.

main CV variables obtained in sitting position, affect-related variables, and medication data of both groups.

## Procedure and clinical/Psychological testing instruments

Participants were instructed to avoid alcohol, smoking, caffeine, and vigorous exercise for 3 hours prior to the experiment, and to refrain from taking any analgesic medications (both non-opioid and opioid) within the 24 hours leading up to the study. Previous studies have validated the current protocol for timing the abstention from alcohol, smoking, caffeine, exercise, and analgesic medications in CV metric research [23, 30, 31]. However, patients were allowed to continue their usual use of antidepressants and anxiolytics, with their potential impact as confounding factors on physiological measures being assessed (see below). The study was conducted in two time frames: morning (10:00 to 13:30) and evening (16:30 to 19:30). A similar proportion of patients and healthy controls participated in the experiment during both time periods. All participants were evaluated in a quiet room with the same light and temperature conditions. First, participants underwent a clinical semi-structured interview to check that they met the inclusion and did not meet the exclusion criteria and to obtain their clinical and demographic data. The following questionnaires (Spanish versions) were administered. The *McGill Pain Questionnaire* (MPQ; [32]) was used to assess clinical pain severity. Habitual pain intensity during the last week was evaluated using the 10-cm-long visual analogue scale (VAS) of the MPQ, anchored on the left by "no pain" and on the right by "pain as bad as it could be", where the participant makes a mark on the line to indicate the intensity of the experienced pain (range: 0–10). Current/present pain intensity (PPI) was measured with the MPQ pain intensity scale, scored as follows: 0, none; 1, mild; 2, discomforting; 3, distressing; 4, horrible; and 5, excruciating. The VAS and PPI were not significantly correlated in FM sufferers in a previous study [33]. Total pain was determined as the global MPQ score, which is a combination of the sensorial, miscellaneous, emotional, and cognitive components of pain (range: 0–167; Cronbach's $\alpha$: 0.74, [34]). The *Beck Depression Inventory-II* (BDI-II; [35]) was used to assess the severity of symptoms of depression (range: 0–63, Cronbach's $\alpha$: 0.95), while the *State-Trait Anxiety Inventory* (STAI; [36]) was used to assess trait anxiety (range: 0–60, Cronbach's $\alpha$: 0.87). The *Fatigue Severity Scale* (FSS; [37]) was used to measure fatigue (range: 9–63, Cronbach's $\alpha$: 0.88), the *Oviedo Quality of Sleep Questionnaire* (OQS; [38]) to assess insomnia severity (range: 9–45, Cronbach's $\alpha$: 0.77), the *Fibromyalgia Impact Questionnaire* (FIQ; [39]) to assess the impact of FM on quality of life (range: 0–100, Cronbach's $\alpha$: 0.91–0.95), and the *Positive and Negative Affect Schedule* (PANAS; [40]) to evaluate positive affect (PA) and negative affect (NA) (range: 10–50, Cronbach's $\alpha$: 0.87 for PA and 0.89 for NA, [41]). Then, participants were seated in an armchair and read the experimental instructions. After placing the electrodes and BP transducers, the "Chronic Pain Autonomic Stress Test" (CPAST) was carried out (see below). During the measurements, participants were instructed not to speak or make sudden movements to avoid interference with the physiological recordings. The psychological and physiological tests were randomly counterbalanced across individuals. All participants provided written informed consent and were fully debriefed. The study protocol was approved by the Ethics Committee of the University of Jaén. The investigation conforms with the principles outlined in the Declaration of Helsinki.

## Active standing and lying posture tests

The "Chronic Pain Autonomic Stress Test" (CPAST) was used as an active clino-orthostatic test adapted for chronic pain research [10, 24]. The procedure included a 5-minute baseline sitting position and a fixed postural test conducted in the following sequence: (1) 1 minute of

standing (initial orthostatic phase, establishing context for subsequent phases); (2) 5 minutes of lying down (clinostatic phase); and (3) 5 minutes of standing (second orthostatic phase). The postural test was conducted twice, with a 20-minute seated rest period between sessions. A second session was incorporated into the original CPAST protocol to enhance the robustness of the test results, mitigating the impact of arousal effects from various between- and within-subject confounding factors that change over time, including adaptation to psychosocial stress and other influences that could affect cardiovascular responses during the clino-orthostatic procedures [10, 42]. Cardiovascular recordings started immediately after the respective postures were taken. Participants were asked to open their eyes while they were standing and during the baseline period and to close them while lying down. The clinostatic and the second orthostatic phases of the procedure in both sessions were analyzed in this study.

## Apparatus and data reduction

A Task Force Monitor (CNSystems, Graz, Austria) was used to record beat-to-beat CV variables. Two electrocardiograms (ECGs) were recorded and bandpass-filtered (0.08 to 150 Hz) by four electrodes applied to the chest, two close to the shoulders, and two at the lower rib cage (Einthoven I and II; for further details, please refer to S Materials.IX in S1 Materials). Four additional electrodes (two at the xiphoidal level in the lateral chest, one at the lower nape, and a ground electrode at the right ankle) were used for impedance cardiography (ICG; for further details, please refer to S Materials.IX in S1 Materials), which was recorded using a 40 kHz current and filtered with a low pass (bandwidth 55.5 Hz) and a high pass (passband * 5 Hz) to remove artifacts and 50 Hz noise. Continuous blood pressure (BP) was taken from the first phalange of the second and third fingers of the right hand (positioned at the level of the heart) and oscillometric BP from the left brachial artery. The device recalibrates continuous finger BP according to brachial artery BP every 60 seconds, without interruption of recording. The ECG and ICG were sampled at 1000 Hz and continuous finger BP at 200 Hz.

Heart rate (HR, beats per min) was derived from ECG by detecting *QRS* complexes using an adaptive threshold decision algorithm implemented in the Task Force Monitor software and computing the time between *R* waves. Systolic, diastolic, and mean blood/arterial pressure levels (SBP, DBP, and MAP, mmHg) were obtained from the continuous finger BP recordings. Stroke volume (SV, ml) was obtained from ICG by using the Kubicek equation [43]; cardiac output (CO, L/min) was calculated as $HR \times SV$; pre-ejection period (PEP, ms) was defined as the period between the *R*-peak in the ECG (ventricular depolarization) and *B* point in the ICG signal (onset of left ventricular ejection); and data on total peripheral or systemic vascular resistance (SVR, dyne*sec*cm$^{-5}$) were directly obtained from the Task Force Monitor, which uses the formula: *80 × (MAP—central venous pressure) / CO*, omitting central venous pressure as it is typically close to 0 mmHg, thereby simplifying the calculation to *SVR = 80 × MAP / CO* [44–46]. These parameters were computed using the algorithms described by Wang et al. and Fortin et al. [47, 48], which reduced ventilation artifacts and motion noise, allowing for better correlations with the thermodilution technique compared to conventional methods [47]. Specifically, for detection of the *B* point in the ICG signal, the algorithm calculates the line between the *dZ/dt* maximum and point where the signal is 30% of the maximum and takes the zero-crossing point of this line as the *B* point. In addition, an online elimination of the influence of breathing is performed, which usually leads to smaller (but more accurate) *dZ/dt* maximums. This effect also shortens the period between the *dZ/dt* maximum and zero-crossing point (zero-crossing occurs later), which renders the obtained PEP values more similar to those obtained from the beginning of the *Q*-wave [47, 49]. Beat-by-beat values for these

variables were converted into second-by-second scores across the full duration of the lying (300 seconds) and standing (300 seconds) postures in both sessions. Before this transformation, beat-to-beat data for all variables were inspected for artifacts. All detected artifacts were corrected by a linear interpolation.

## Sample size calculations

Using means and standard deviations of CV responses to active standing from a previous study [50], error probability ($\alpha$) of 0.05 and power ($1$-$\beta$) of 0.80, and using these values for sample size calculation, a sample of 22 participants per group appeared optimal.

## Statistical analysis

All continuous data were treated as sets of curves to capture in time the correct shapes of passively altered hydraulic (by body water or blood translocations between central and peripheral organs and tissues) and actively regulated compensating hemodynamic (by various basic reflexes and regional reactions) moment-to-moment responses of the CV system to the changes between clino- and ortho-static postures as gravitation-related challenges [26]. Changes of the postures were used as events synchronizing hemodynamic and CV fluctuations among participants for better detection of grouping effects on the fluctuations with their different (longer or shorter) periods, (higher or lower) rates, (higher or lower) magnitudes, and amplitudes determined or modulated by stochastic processes (treated as individual random effects) and a pain or pain-free status (treated as a grouping factor fixed effect) with other covariates such as age, height, weight, and body mass index (treated as additional fixed effects). Data-driven adaptive techniques, as detailed below, were employed for data analysis to account for the expected high leverage of individual variability in responses. These techniques considered the modulating effects of periodic activities (e.g., breathing-heart and breathing-vascular couplings) and aperiodic activities (e.g., hydrostatic and muscle-heart couplings) relevant to posture changes, while controlling for irrelevant physiological reactions and equipment faults, treating them as internal stochastic outliers and external artifacts to be disregarded as noise [26, 27, 51, 52].

   Since time of these task-relevant and irrelevant responses may non-linearly fluctuate within specific ranges between subjects and for repeated challenges, non-linear models, as extensions of multivariate analysis techniques to the cases where the data are a set of curves instead of vectors, were used to draw conclusions. Indeed, modeling non-linear responses as if they were linear not only results in inaccurate predictions but also in structured errors that, for Gaussian models for instance, may show heteroscedasticity, and thus their reported significances may be considered unreliable. However, despite their continuous nature, the hemodynamic and CV data curves in the present study were represented as finite sets of second-by-second sampling points that could unevenly measure a superimposition of fast and slow physiologically activities at different periods of the clino-orthostatic responses. To address this, non-parametric approximation techniques were applied using generalized additive model (GAM) and generalized additive mixed model (GAMM) functions from the *R* package *mgcv* [27, 51–53], with interpolation of finite-dimensional spaces generated by basis functions (*bs*) and automated data-driven model selection, as detailed below. This approach was used to reconstruct the functional forms of each hemodynamic and CV reactivity curve from a finite set of discrete sampling points, accounting for non-linear effects and outliers. Hence, this non-linear methodology was designed to overcome the limitations of multiple individual linear analyses, which treat data as vectors across predefined short- or long-term periods during clino-orthostatic challenges, each with fixed temporal resolutions for hemodynamic responses. Consequently,

this automated data-driven approach during a singular analysis was anticipated to detect any hemodynamic or CV responses exhibiting varying velocities, durations, periodicities, and trends, potentially sensitive to the effects induced by clinical pain conditions such as FM syndrome.

The GAMM function was initially employed for this data modeling by applying *B*-splines as the ideal basis system for data mixed by non-periodic and periodic modulations observed with noise [27, 53]. A penalized smoothing spline algorithm (*P*-splines, *ps*) with penalized quasi-likelihood (PQL) computations as an approximation and model validation technique with 'performance-oriented iteration' was chosen as a method to balance between under- and over-fitting (i.e., between the degrees of freedom and the goodness of fit) of integrated mixed models in adaptively optimizing the number and the position of knots (*k*) or curve probable turning points (i.e., for choosing the smoothing parameters or lambdas [$\lambda$]) in reconstructing the curve smoothly [27]. This (*P*-splines) kind of penalized smoothers is considered to work better with a relatively large number of knots as in the hemodynamic and CV data modulated by different physiological processes [26, 54]. In these smoothing spline algorithms, the adaptiveness to data heterogeneity is achieved by automatic modeling of the smoothing parameter ($\lambda$) as a penalty function of the independent variable minimizing the problem with a new penalty function using a PQL method [27, 51, 53]. Thus, the method's principle involves using a large number of knots, with $\lambda$ controlling the degree of smoothness based on PQL computations for model validation. As a result, these modeling algorithms do not require predefining the dynamics of the processes, such as linear or non-linear with specific periodic or non-periodic fluctuations, because the shapes of their activity and reactivity are determined by the data itself [54]. The models obtained by GAMM were further confirmed by similar nonparametric regression techniques included in the GAM package using another data driven validation criterium (generalized cross validation, GCV) with adjustment of the model edf (effective degrees of freedom or curve 'actual' turning points) by gamma ($\gamma$) below 1 as a multiplier for adaptively modeling the smoothing parameter ($\lambda$) to data heterogeneity [26, 27, 52, 53]. Values of edf in GAMM and GAM can be used as estimates of principal or main periodic and aperiodic fluctuations in continuous temporal dynamics (i.e., including fluctuations in steady-state and transient components) of multiple-degree-of-freedom non-linear models such as the activity of the CV system. As such, this approach could be extended to evaluate dynamics of periodic systems similar to other frequency and time-domain techniques such as the Fourier transform and the autoregression analysis. Thus, this multi-degree-of-freedom modeling may be another approach in transformations (conversions) of discrete-time signals (e.g., a sequence of real CV numbers sampled in seconds) into a time- or frequency-domain representation.

Adaptive model fitting in both the GAMM and GAM functions was specified by (i) setting the basis (*bs*) order and penalty to prevent overfitting, addressing under-smoothing and managing outliers (in this study, the second order for both was used; represented by the *m* value in the equations example below) and (ii) selecting the maximum possible number of knots or basis dimensions (represented by the *k* value in the equations example below), using a full search algorithm to prevent underfitting and counteract over-smoothing. In the present study, this number of knots (*k*) was fixed to 240 to cover the theoretically predicted range (from slow to fast) of modulating effects on CV measures from various physiological factors during the 600 seconds of challenging periods. Qualities of the model fitting were confirmed by a so-called *k-index* being above 1.0 combined with a high *p*-value and actual edf being not too close to *k'* as the maximum possible edf calculated by the *gam.check()* function included in the *mgcv* package [53]. Autoregressive correction of hemodynamic and CV data was not applied in the formulas of the functions as it further increased under-fitting (i.e., over-smoothing) and removed fast hemodynamic and CV fluctuations modulating responses. Since random effects

in the study were part of the experimental design, but not a focused variable (i.e., not a research question of specific subject effects), the contribution of the subject specific random effects was not included in the models when estimating and analyzing fixed effects by respective modeling techniques.

Models that produced similar results in both the GAMM and GAM functions, using data-driven validation criteria (PQL for GAMM and GCV for GAM), were selected to plot the shapes of hemodynamic and cardiovascular processes for each group, highlighting and comparing their differences. These plots were generated using the *ggplot()* function from the *ggplot2* package and *plot_diff()* from the *itsadug* package in *R* [55, 56]. The effects were reported with 95% CIs for the model fits. Using these drawing functions, between-group differences in the curves of hemodynamic and CV measures were depicted in two groups of figures: (i) with generation of approximate 95% confidence intervals (CIs) as a less conservative approach for indicating time periods with significant time effects on hemodynamic and CV curves averaged per groups without adjustment for covariates and (ii) with generation of simultaneous 95% CIs with 10000 samples in posterior simulations as a more conservative approach for multiple comparisons in indicating time periods, presented with significance of the differences before and after control for covariates in those cases when the covariates showed significant linear effects. In these cases, directions of significant effects were indicated by position of hemodynamic or CV values with 95% CI above or below the zero line without its inclusion (marked in the respective figures by horizontal bold lines on the *X*-axis, bounded by vertical dashed lines, corresponding to the relevant points along the hemodynamic or CV difference curves).

Both the Akaike (AIC) and Bayesian (BIC) Information Criteria were additionally used to confirm the best fitting models among alternatives with different numbers of basis dimensions, interaction terms, and covariates [57]. The BIC metric, developed as a goodness of fit measure, was preferred in those cases where more than three candidate structures were in mixed effects models for large continuous data since it applied a much larger penalty for complex models to accurately infer the "true distribution" of the current data and where the AIC metric, developed as a predictive accuracy measure, could predict a wrong future data structure from the current data [58].

Thus, the statistical models were proposed to reach the main objective of the study—to assess the impact of belonging to a group of female patients with FM syndrome in contrast to a CT group of healthy women, i.e., a grouping according to presence/absence of the chronic pain, on curves of hemodynamic (re)activity continuously over the whole clino-orthostatic challenges assessed by different CV measures: SV, CO, SVR, PEP, HR, SBP, and DBP. Therefore, the focus was on identifying distinct hemodynamic and CV response patterns characterized by significant differences in timing, velocity, duration, shape, periodicity, and trend of the curves for each group, treated as an independent variable. An interaction of the Group factor with time, as predictors of hemodynamic and CV changes, was modeled in the following form (presented for SBP as an example of continuous non-linear analysis of the curves, with additional Group, Session, and Posture simple and interaction effects on CV means, assessed as linear):

*SBP ~ Session \* Group + Posture \* Group + s(Time, k = 240, m = 2, bs = "ps", by = Group)*

The smooth *s()* term specifies a 2nd-order *P*-spline basis (cubic spline) with a 2nd-order difference penalty on the coefficients, indicated by the arguments *bs = "ps"* and *m = 2*, with the number of knots (*k*) set to 240 for each Group, using the *by* argument. Further details on the variables used in these statistical models and the rationale behind the chosen values are provided earlier. Simple effects for Group, Session, Posture, and their interactions (Group \* Session, Posture \* Group) were analyzed using linear regression models with the *F*-statistic, as

part of the same *R* command for non-linear effects within the *mgcv* package. The same hemo-dynamic and CV measures were evaluated both before and after adjusting for body mass index (BMI), weight, height, and age, which were included as covariates either individually or together. For example, the adjustment for BMI is presented in the following form:

*SBP ~ Session * Group + Posture * Group + s(Time, k = 240, m = 2, bs = "ps", by = Group) + BMI*

The potential impact of medications was analyzed for each physiological variable in the FM group by including a factor that stratified the group based on medication use (separately for antidepressants, anxiolytics, non-opioid analgesics, and opiates) in the models to assess both main and interaction effects. Since none of these medication factors showed significant effects, they were excluded from further analyses.

An interaction between the Session factor and time, as predictors of CV changes, was also modeled using a similar approach (see details above) and was represented in the following form:

*SBP ~ s(Time, k = 240, m = 2, bs = "ps", by = Session)*

The results from these models, shown in S Materials.I in S1 Materials, highlight within-subject reconditioning and between-session mild dehydration effects on temporal hemodynamics during recurrent clino-orthostatic challenges, with approximate 95% CIs.

All parameter estimates are expressed as non-standardized (*b*) regression coefficients with standard errors (SE). Two-tailed *p*-values $< .05$ were regarded as statistically significant in the linear and non-linear models. Effects in Table 1 were evaluated using either a two-tailed Student's *t*-test for group differences in numerically scaled dependent variables or a non-parametric chi-squared ($\chi^2$) test to assess relationships between groups and nominal or ordinal variables [59, 60]. To evaluate additional linear effects, the percentile bootstrap procedure (with 5000 bootstrap samples) was employed in some linear regression analyses to generate non-parametric 95% CIs for the regression coefficients [61, 62]. Relationships in the models were considered statistically significant when the CIs did not contain zero, indicating the results were unlikely due to chance. Bootstrapping is widely used as a flexible and robust non-parametric method for statistical inference, particularly when parametric assumptions (like normal distribution) are uncertain or not applicable, a common scenario in biological and psychological research [61]. Thus, in the present study, between-group differences were used to explore effects of FM as a chronic pain disorder in general, i.e., as a predictive factor represented by averaged means of pain severity estimates in the current patient sample (Table 1) and 10.21 (6.49) of chronicity in years (SD) from the FM diagnosis, on the temporal clino-orthostatic hemodynamics, as a pain-o-metry based physiological process in the syndrome compared to the group of healthy individuals.

## Results

### Models for SV

With 44.6% of explained variance in both GAMM and GAM, significant simple Session, Group, and Posture linear effects were obtained for SV with its higher mean during session 1 compared to session 2 (*b*[SE] = -2.47 [0.10], *t* = -23.94, *p* < 0.0001; here and further presented as in GAMM), in the FM group compared to the CT group (*b*[SE] = -13.33 [3.47], *t* = -3.84, *p* = 0.0001), during lying compared to standing (*b*[SE] = -13.60 [0.82], *t* = -16.52, *p* < 0.0001), and in those with higher height (*b*[SE] = 104.26 [23.73], *t* = 4.39, *p* < 0.0001) and lower age (*b*[SE] = -0.57 [0.19], *t* = -2.98, *p* = 0.0029). Significant interaction Session * Group and Posture * Group linear effects were obtained for SV with its unchanging low level from session 1 to session 2 in the CT group compared to the FM group for which it decreased from a higher level at

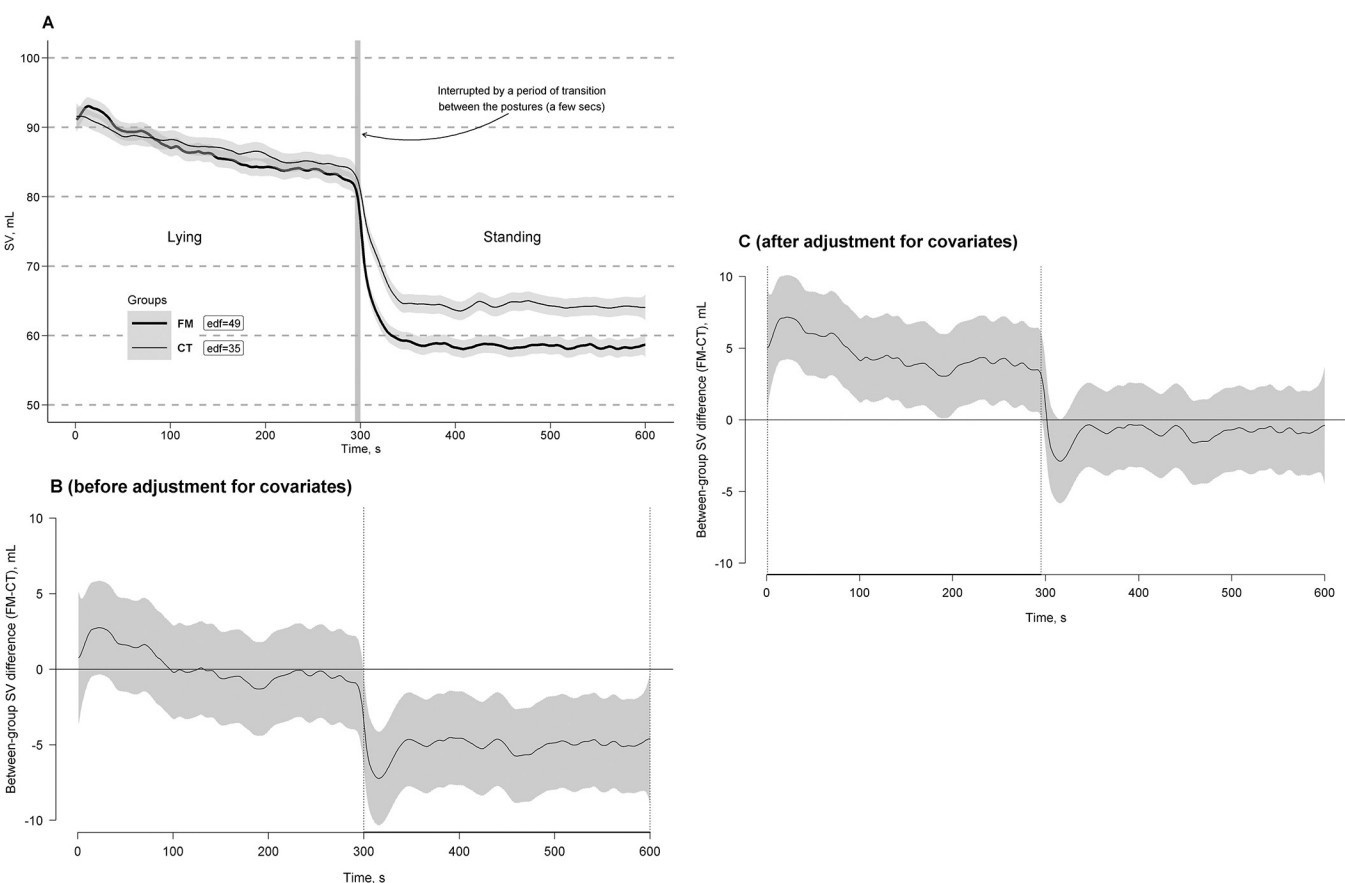

**Fig 1. Estimated time effects on stroke volume (SV) curves during the clino- and ortho-static challenges.** The effects are averaged for the fibromyalgia (FM, bold line) and control (CT, thin line) groups, with approximate 95% confidence intervals (gray shading around the curves) and effective degrees of freedom or curve 'actual' turning points (edf) (A). Time windows showing significant between-group differences (FM-CT) are indicated by the position of SV values with simultaneous 95% confidence intervals (gray shading around the curves) either above or below the zero line, without crossing it, and depicted as horizontal bold-line projections on the Time ($X$) axis, bounded by vertical dashed lines, corresponding to relevant points along the SV difference curves, both before (B) and after (C) adjustment for covariates (height and age).

session 1 to a similarly low level at session 2 ($b$[SE] = 2.31 [0.15], $t$ = 15.34, $p < 0.0001$), and with a smaller decrease of SV from its lower level in the CT group compared to a deeper drop of SV from its higher level in the FM group in response to the transition from a lying to standing posture ($b$[SE] = 6.07 [1.22], $t$ = 4.99, $p < 0.0001$). A significant Group effect was obtained for SV fluctuation during the clino-orthostatic challenge using $P$-spline adaptive smoothing models with different curve reconstructing criteria, defining "actual" or obtained numbers of curve turning points, i.e., edf values, equal to 49 for the FM group and 35 for the CT group ($F$[18.29, 96859.33] = 87.47, $p < 0.0001$) in GAMM and equal to 49 for the FM group and 33 for the CT group ($F$[21.85, 96855.07] = 24.11, $p < 0.0001$) in GAM, i.e., with a slightly higher frequency of SV fluctuation in the patients with chronic pain (i.e., about 5 cycles of SV fluctuation per min during the challenges) compared to the healthy participants (i.e., about 3.3–3.5 cycles of SV fluctuation per min).

Fig 1A shows the estimated time effects on SV curves averaged per the FM and CT groups with approximate 95% CIs. As can be seen, responses to the clino- and ortho-static challenges (i.e., clino- and ortho-static effects on the SV steady states) were similar across the groups in general trends but different in amplitudes: a slow step-wise mean decrease of SV as its main

trend in both groups during the clinostatic state after its common rapid transition-related rise in response to the transition from standing to the lying posture; then rapid SV drops (stronger in the FM group) in response to the transition from lying to standing were followed by their stable horizontal trends of SV curves in both groups persisting through the whole standing posture as their common steady state. Almost similarly fast SV curve fluctuations were detected to be superimposed on these trends and steady states in both groups.

Time windows of significant between-group differences of the curves with simultaneous 95% CIs before and after adjustment for covariates (height and age) were depicted in Fig 1B and 1C. Differences of unadjusted SV curves between the groups became significant only after standing up, and this difference persisted through the entire orthostatic challenge (time window: 300–600 s) with a much lower SV curve level in the FM group compared to the CT group (Fig 1B). The greatest number of significant between-group differences in the adjusted SV curves were obtained just after lying down, persisting through the entire clinostatic position (time window: 1–295 s), and with a much higher SV curve level in the FM group compared to the CT group (Fig 1C). Immediately after standing up, the SV curve levels became close to each other with no significant differences between the groups.

## Models for HR

With 23.3% of explained variance in both GAMM and GAM, significant main Session, Group, and Posture linear effects were obtained for HR with its higher mean during session 1 compared to session 2 ($b$[SE] = -1.74 [0.04], $t$ = -39.98, p < 0.0001; here and further presented as in GAMM), in the FM group compared to the CT group ($b$[SE] = -9.37 [2.31], $t$ = -4.07, $p < 0.0001$) and during standing compared to lying ($b$[SE] = 14.71 [0.52], $t$ = 28.10, $p < 0.0001$). Significant interaction Session * Group and Posture * Group linear effects were obtained for HR with its lesser decrease from session 1 to session 2 in the CT group compared to the FM group for which it decreased from a higher level at session 1 to a similarly low level at session 2 ($b$[SE] = 0.23 [0.06], $t$ = 3.55, $p = 0.0004$), and with a stronger HR rise from its higher level in the FM group compared to a smaller HR rise from its lower level in the CT group in response to the transition from a lying to standing posture ($b$[SE] = -0.75 [0.06], $t$ = -11.66, $p < 0.0001$). A significant Group effect was obtained for HR fluctuation during the clino-orthostatic challenge using $P$-spline adaptive smoothing models with different curve reconstructing criteria, defining "actual" or obtained numbers of curve turning points, i.e., edf values, equal to 158 for the FM group and 181 for the CT group ($F$[52.62, 96684.30] = 1.84, $p < 0.0001$) in GAMM and edf values equal to 154 for the FM group and 183 for the CT group ($F$[34.15, 96697.76] = 0.45, $p < 0.0001$) in GAM, i.e., with HR fluctuation with lower frequency in the patients with chronic pain (i.e., about 15.4–15.8 cycles of HR fluctuation per min during the challenges) compared to the healthy participants (i.e., about 18.1–18.3 cycles of HR fluctuation per min). A significant Group effect using $P$-spline adaptive smoothing models with different curve reconstructing criteria was also obtained for HR fluctuation around 0.1 Hz with obtained numbers of curve turning points, i.e., edf values, equal to 64 for the FM group and 65 for the CT group ($F$[37.17, 96832.87] = 6.02, $p < 0.0001$) in GAMM and equal to 65 for the FM group and 66 for the CT group ($F$[26.86, 96842.64] = 3.33, p < 0.0001) in GAM, i.e., with HR fluctuation with almost similar frequency in the patients with chronic pain and healthy participants (i.e., about 6.4–6.5 and 6.5–6.6 cycles of HR fluctuation per min, respectively).

Fig 2A and 2B show the estimated time effects on HR curves with high and low HR frequency fluctuations averaged per the FM and CT groups with approximate 95% CIs. As can be seen, response to the clino- and ortho-static challenges (i.e., clino- and ortho-static effects on the HR steady states) were similar across the groups in general trends but different in fast

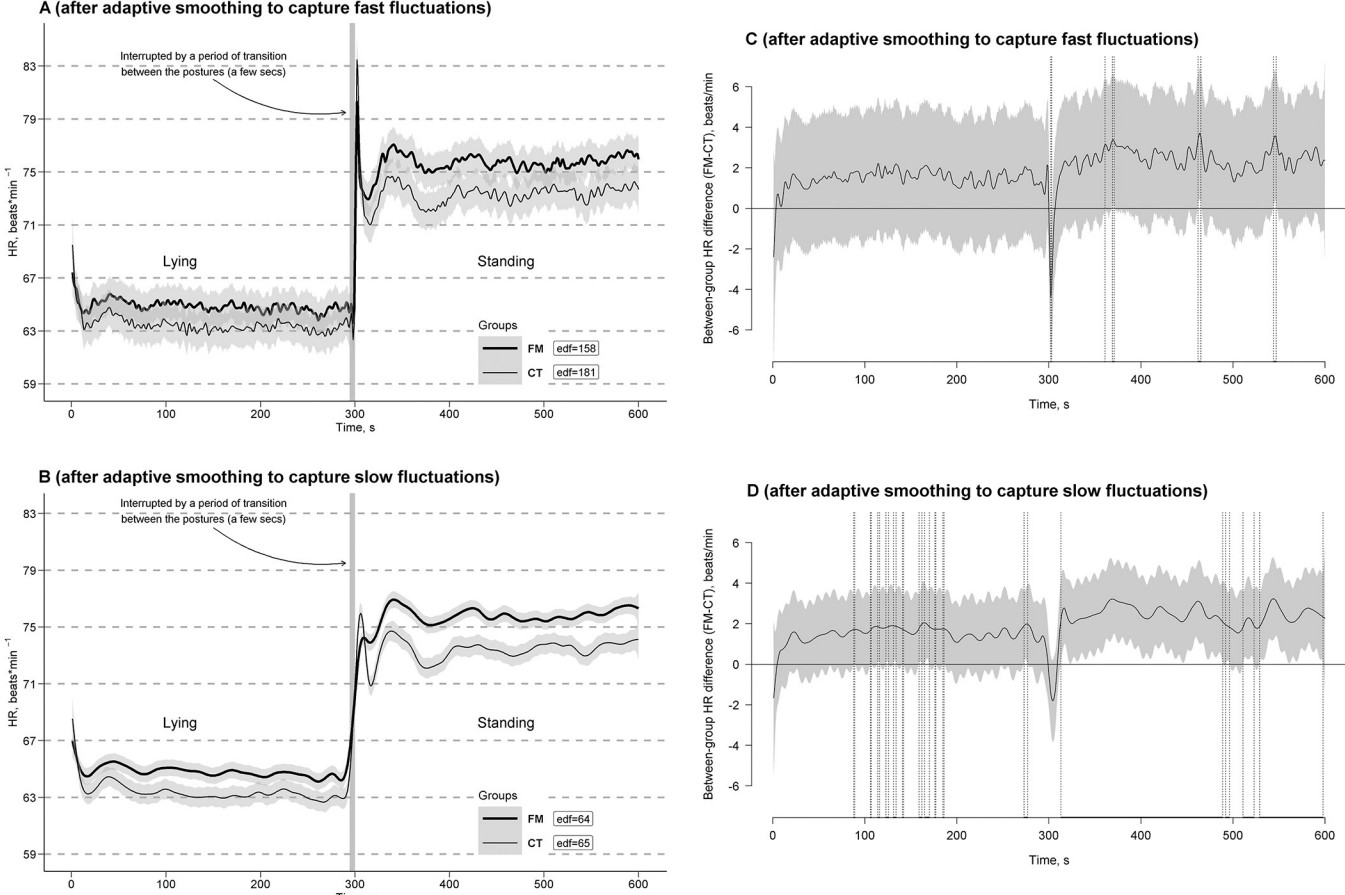

**Fig 2. Estimated time effects on heart rate (HR) curves during the clino- and ortho-static challenges.** The effects are averaged for the fibromyalgia (FM, bold line) and control (CT, thin line) groups, with approximate 95% confidence intervals (gray shading around the curves) and effective degrees of freedom or curve 'actual' turning points (edf) identified after adaptive smoothing to detect fast (A) or slow (B) fluctuations. Time windows showing significant between-group differences (FM-CT) are indicated by the position of HR values with simultaneous 95% confidence intervals (gray shading around the curves) either above or below the zero line, without crossing it, and depicted as horizontal bold-line projections on the Time (*X*) axis, bounded by vertical dashed lines, corresponding to relevant points along the HR difference curves for both fast (C) and slow (D) fluctuations.

fluctuations. There was a mostly stable mean of HR as its main vector in both groups during the clinostatic state after their common rapid transition-related drop from a standing to lying posture.

Also seen was a transient (rapid and short) primary HR overshooting rise immediately after the transition from lying to standing (higher in the CT group compared to the FM group) followed by HR curve drops to levels higher than during the lying posture with subsequent secondary rises and drops of lower amplitudes. This was followed by a relative stabilization of HR in curves at levels higher than during the lying posture in both groups (with a higher HR curve level in the FM group compared to the CT group), which persisted by the end of the standing posture as a steady state during the challenge. Fast and slow HR curve fluctuations were detected to be superimposed on these posture-related steady states in both groups.

Time windows of significant between-group differences of the curves with simultaneous 95% CIs after adaptive smoothing for detecting either fast or slow fluctuations were depicted in Fig 2C and 2D, respectively. Curves adaptively smoothed for detecting fast fluctuations showed a main significant between-group difference for the transient HR overshooting rise

observed shortly after standing (time window: 300–303 s). Other significant differences were observed in very narrow (with 1 to 3 second periods) time windows randomly and rarely distributed across the whole standing posture, where were likely the result of overlapping of CIs of HR between the FM and CT groups due to the interference of fast HR changes fluctuating with frequencies non-synchronized between the groups (15.4–15.8 and 18.1–18.3 cycles per min) (Fig 2C). When the impact of fast HR fluctuations was removed from the effects by adaptive curves smoothing for detecting slow fluctuations with frequencies synchronized between the groups (6.4–6.5 and 6.5–6.6 cycles per min), between-group difference was observed to be mainly significant for stable HR curve vectors during the standing posture (time window: 313–598 s) with a higher HR curve level in the FM group compared to the CT group interrupted by short random non-significant periods. This difference was largely nonsignificant during the lying posture, as the 95% CIs included zero throughout most of the period, with only brief intervals of significance observed between 90 and 200 ms (Fig 2D).

## Models for CO

With 31.4% of explained variance in both GAMM and GAM, significant main Session, Group, and Posture linear effects were obtained for CO with its higher mean during session 1 compared to session 2 ($b$[SE] = -0.28 [0.01], $t$ = -39.65, $p < 0.0001$; here and further presented as in GAMM), in the FM group compared to the CT group ($b$[SE] = -1.86 [0.26], $t$ = -7.06, $p < 0.0001$), during standing compared to lying ($b$[SE] = 0.30 [0.07], $t$ = 4.48, $p < 0.0001$), and in those with higher height ($b$[SE] = 7.25 [1.67], $t$ = 4.34, $p < 0.0001$) and lower age ($b$[SE] = -0.04 [0.01], $t$ = -2.89, $p = 0.0039$). Significant interaction Session * Group and Posture * Group linear effects were obtained for CO with its small decrease from session 1 to session 2 in the CT group compared to the FM group with a stronger decrease from a higher level at session 1 to a similarly low level at session 2 ($b$[SE] = 0.14 [0.01], $t$ = 13.51, $p < 0.0001$), and with a stronger drop of CO from its higher level in the FM group compared to its smaller drop from its lower level in the CT group in response to the transition from a lying to standing posture ($b$[SE] = 1.01 [0.11], $t$ = 9.32, $p < 0.0001$). A significant Group effect was obtained for CO fluctuation during the clino-orthostatic challenge using $P$-spline adaptive smoothing models with different curve reconstructing criteria, defining "actual" or obtained numbers of curve turning points, i.e., edf values, equal to 37 for the FM group and 67 for the CT group ($F$[28.29, 96837.74] = 50.84, $p < 0.0001$) in GAMM and equal to 33 for the FM group and 74 for the CT group ($F$[24.96, 96838.28] = 17.41, $p < 0.0001$) in GAM, i.e., with CO fluctuation with twice lower frequency in the FM group (i.e., about 3.3–3.7 cycles of CO fluctuation per min during the challenges) compared to the CT group (i.e., about 6.7–7.4 cycles of CO fluctuation per min).

Fig 3A shows the estimated time effects on CO curves averaged per the FM and CT groups with approximate 95% CIs. As can be seen, response to the clino- and ortho-static challenges (i.e., clino- and ortho-static effects on the CO steady states) were similar across the groups in general vector trends but different in response amplitudes and fast fluctuations: the slow stepwise mean decrease of CO as its main trend in both groups during a clinostatic state after its common rapid rise in response to the transition from a standing to lying posture; rapid CO drops in response to the transition from lying to standing (deeper in the FM group compared to the CT group) followed by a relatively stable horizontal vector of CO in both groups persisting by the end of the standing posture as their common steady state during the challenge. However, in contrast to the FM group, an additional transient (rapid and short) CO overshooting rise was observed in the CT group just after standing. Fast CO curve fluctuations were detected to be superimposed on these trends and steady states in both groups.

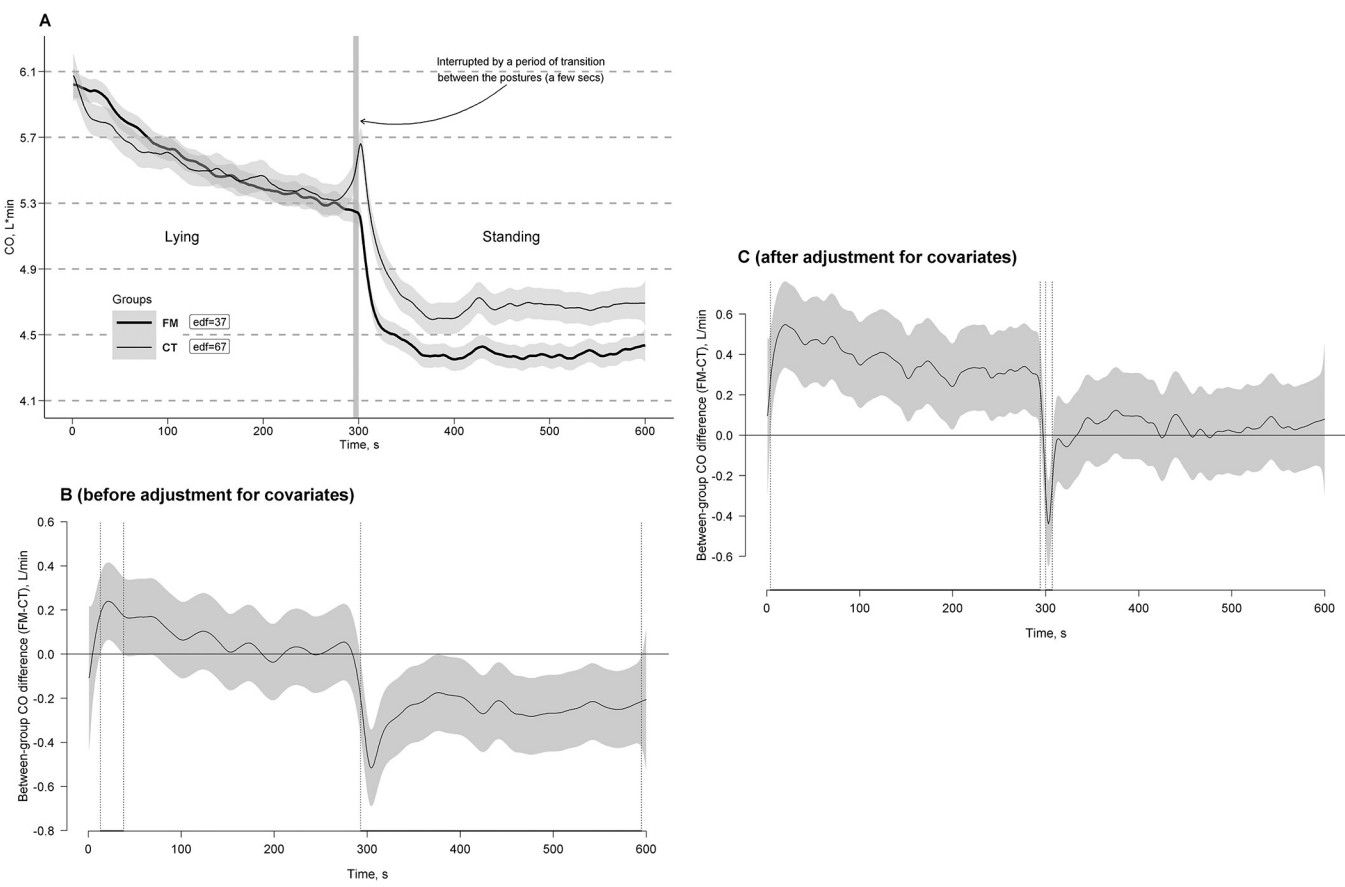

**Fig 3. Estimated time effects on cardiac output (CO) curves during the clino- and ortho-static challenges.** The effects are averaged for the fibromyalgia (FM, bold line) and control (CT, thin line) groups, with approximate 95% confidence intervals (gray shading around the curves) and effective degrees of freedom or curve 'actual' turning points (edf) (A). Time windows showing significant between-group differences (FM-CT) are indicated by the position of CO values with simultaneous 95% confidence intervals (gray shading around the curves) either above or below the zero line, without crossing it, and depicted as horizontal bold-line projections on the Time (*X*) axis, bounded by vertical dashed lines, corresponding to relevant points along the CO difference curves, both before (B) and after (C) adjustment for covariates (height and age).

Time windows of significant between-group differences of the curves with simultaneous 95% CIs before and after adjustment for covariates (height and age) were depicted in Fig 3B and 3C, respectively. A short time window with between-group significant differences of unadjusted CO curves was observed shortly after lying down (time window: 13–38 s) with no significant between-group differences through the rest of the clinostatic challenge (Fig 3B). The unadjusted CO curves had again significant between-group differences immediately after standing with their further persistence through the whole period of the orthostatic challenge (time window: 300–595 s) with a much lower drop of the CO curve level in the FM group compared to the CT group. However, the greatest number of significant between-group differences in the adjusted CO curves were obtained just after lying down and persisted throughout the entirety of the clinostatic challenge (time window: 4–294 s) with a much higher CO curve rise in the FM group compared to the CT group, as well as immediately after standing (time window: 300–307 s) with a much higher level of the CO curve in the CT group (Fig 3C). Later on, the CO curves of the groups became close to each other with no significant between-group differences.

## Models for PEP

With 25.5% of explained variance in both GAMM and GAM, significant main Session and Posture linear effects were obtained for PEP with its higher mean during session 2 compared to session 1 ($b$[SE] = 1.11 [0.10], $t$ = 11.38, $p$ < 0.0001; here and further presented as in GAMM) and during standing compared to lying ($b$[SE] = 7.95 [0.62], $t$ = 12.78, $p$ < 0.0001). Significant interaction Session * Group and Posture * Group linear effects were obtained for PEP with its stronger increase from session 1 to session 2 in the CT group compared to the FM group with its smaller increase from its higher level at session 1 to a similarly low level at session 2 ($b$[SE] = 0.65 [0.14], $t$ = 4.57, $p$ < 0.0001), and with a stronger PEP rise in the CT group compared to a smaller PEP rise in the FM group from similar levels in response to the transition from a lying to standing posture ($b$[SE] = 1.82 [0.14], $t$ = 12.61, $p$ < 0.0001). A significant Group effect was obtained for PEP fluctuation during the clino-orthostatic challenge using *P*-spline adaptive smoothing models with different curve reconstructing criteria, defining "actual" or obtained numbers of curve turning points, i.e., edf values, equal to 62 for the FM group and 60 for the CT group ($F$[27.49, 96837.45] = 1.62, $p$ < 0.0001) in GAMM and equal to 69 for the FM group and 55 for the CT group ($F$[41.54, 968807.29] = 0.96, $p$ < 0.0001) in GAM, i.e., with PEP fluctuation with slightly higher frequency in the FM group (i.e., about 6.2–6.9 cycles of PEP fluctuation per min during the challenges) compared to the CT group (i.e., about 5.5–6.0 cycles of PEP fluctuation per min).

Fig 4A shows the estimated time effects on PEP curves averaged per the FM and CT groups with approximate 95% CIs. As can be seen, response to the clino- and ortho-static challenges (i.e., clino- and ortho-static effects on the PEP steady states) were close across the groups in general trends of their curves but slightly different in response amplitudes: the slow step-wise increase of the PEP curve level as its main vector of trend in the FM group in contrast to a stable level of the PEP curve in the CT group during the clinostatic challenge after their common rapid drop in response to the transition from a standing to lying posture. A rapid PEP rise in response to the transition from lying to standing was found in both groups; however, the orthostatic PEP rise was more curvilinear with a smaller amplitude in the FM group compared to straighter with a higher amplitude and a short (few seconds) delay in the CT group. The rise was followed by relatively stable horizontal trends of PEP curves in both groups (with a higher PEP curve level in the CT group compared to the FM group), which persisted through the whole period of the standing posture as their common steady state. Fast PEP curve fluctuations were detected to be superimposed on these vector trends and steady states in both groups.

Time windows of significant between-group differences of the curves with simultaneous 95% CIs were depicted in Fig 4B. A few significant between-group differences in the PEP curve were observed just after standing, though they did not persist through the entire orthostatic challenge (time window: 303–314 s), indicating a faster PEP curve rise just after standing in the FM group compared to the CT group. Later on, the PEP curve level became higher in the CT group compared to the FM group but without significant between-group differences as was indicated by the more conservative statistical approach.

The analysis of the correspondence between SV, an indicator of central blood volume, and PEP, an indicator of cardiac contractility, revealed a relatively lower compensation for the SV drop by PEP prolongation throughout standing in the FM group (*SV×PEP* = 7.19[1.36] mL*sec) compared to the CT group (*SV×PEP* = 8.03[1.69] mL*sec). This indicates a statistically significant between-group difference in the adherence to the Frank-Starling Law ($b$[SE] = -0.84[0.34], t = -2.47, $p$ = 0.016, bootstrap 95% CI = -1.52 to -0.18; see Figs 1 and 4). Conversely, during the lying posture, SV was comparably balanced by PEP in both groups (*SV×PEP* = 9.13[2.59] in the FM group and 9.21[2.75] in the CT group), with no significant

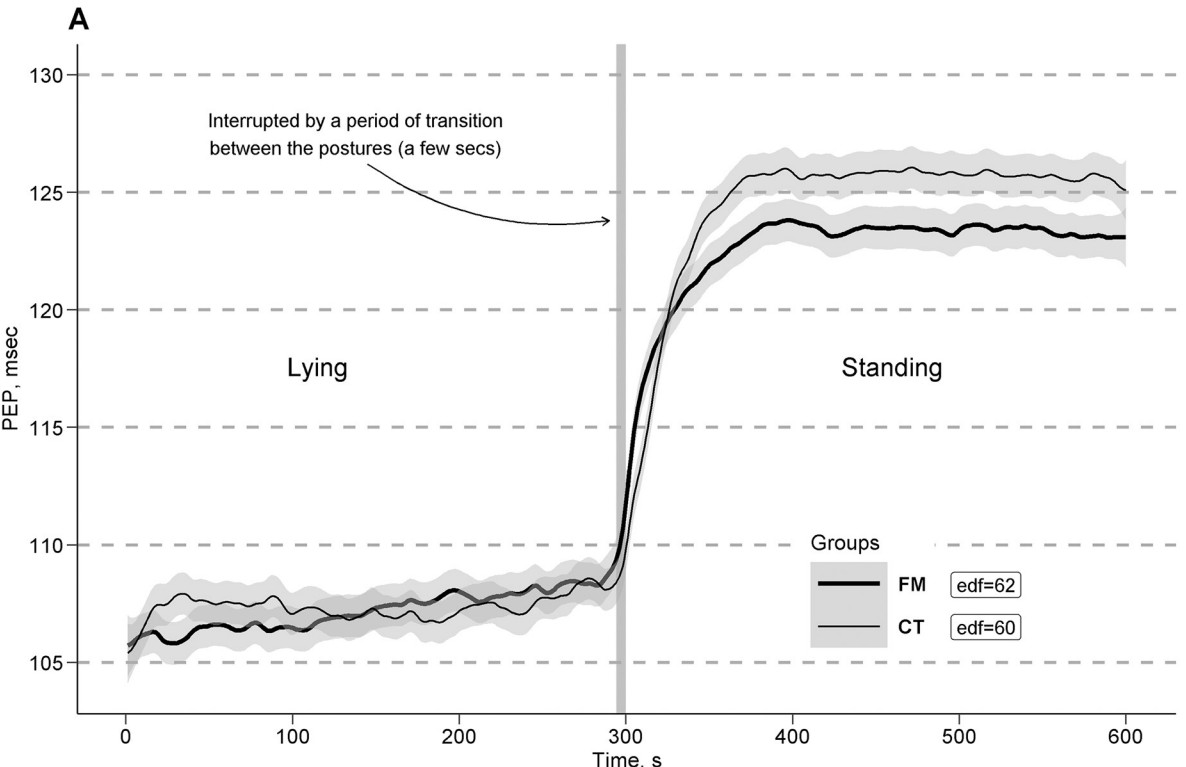

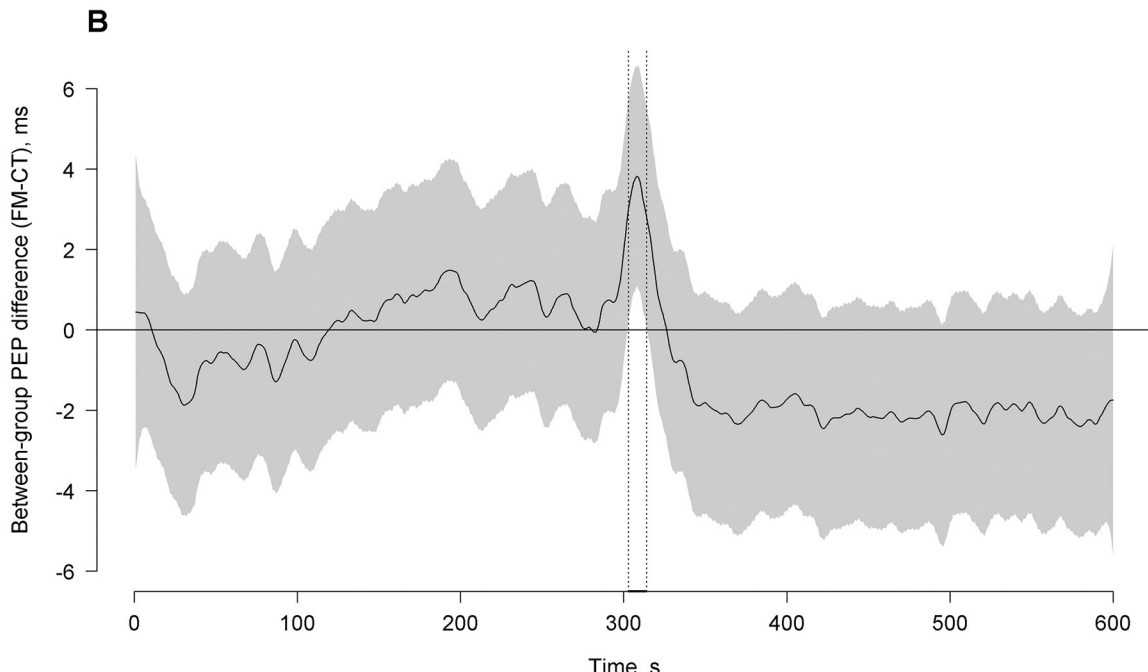

**Fig 4. Estimated time effects on pre-ejection period (PEP) curves during the clino- and ortho-static challenges.** The effects are averaged for the fibromyalgia (FM, bold line) and control (CT, thin line) groups, with approximate 95% confidence intervals (gray shading around the curves) and effective degrees of freedom or curve 'actual' turning points (edf) (A). Time windows showing significant between-group differences (FM-CT) are indicated by the position of PEP values with simultaneous 95% confidence intervals (gray shading around the curves) either above or below the zero line, without crossing it, and depicted as horizontal bold-line projections on the Time (*X*) axis, bounded by vertical dashed lines, corresponding to relevant points along the PEP difference curves (B).

between-group difference in the adherence to the Frank-Starling Law ($b$[SE] = -0.08[0.59], t = -0.14, $p$ = 0.89, bootstrap 95% CI = -1.28 to 1.11).

## Models for SVR

With 21.4% of explained variance in both GAMM and GAM, significant main Session and Posture linear effects were obtained for SVR with its higher mean during session 2 compared to session 1 ($b$[SE] = 72.20 [1.98], $t$ = 36.54, $p$ < 0.0001; here and further presented as in GAMM), during standing compared to lying ($b$[SE] = 301.74 [2.05], $t$ = 147.57, $p$ < 0.0001), and in those with lower height ($b$[SE] = -2076.68 [515.57], $t$ = -4.03, $p$ < 0.0001). Significant interaction Session * Group and Posture * Group linear effects were obtained for SVR with its lesser increase from session 1 to session 2 from a higher level in the CT group compared to the FM group ($b$[SE] = -10.83 [2.89], $t$ = -3.75, $p$ = 0.0002) and with a higher SVR rise from its lower level in the FM group compared to a lower SVR rise from its higher level in the CT group in response to the transition from a lying to standing posture ($b$[SE] = -186.03 [28.20], $t$ = -6.60, $p$ < 0.0001). A significant Group effect was obtained for SVR fluctuation during the clino-orthostatic challenge using $P$-spline adaptive smoothing models with different curve reconstructing criteria, defining "actual" or obtained numbers of curve turning points, i.e., edf values, equal to 40 for the FM group and 68 for the CT group ($F$[34.24, 96823.47] = 85.14, $p$ < 0.0001) in GAMM and equal to 39 for the FM group and 72 for the CT group ($F$[28.49, 96823.08] = 26.96, $p$ < 0.0001) in GAM, i.e., with SVR fluctuation with about 1.7 times lower frequency in the FM group (i.e., about 3.9–4.0 cycles of SVR fluctuation per min during the challenges) compared to the CT group (i.e., about 6.8–7.2 cycles of SVR fluctuation per min).

Fig 5A shows the estimated time effects on SVR curves averaged per the FM and CT groups with approximate 95% CIs. As can be seen, general trends of SVR curves in response to the clino- and ortho-static challenges (i.e., clino- and ortho-static effects on the SVR steady states) were different across the groups and mimicked or repeated the trends of the DBP curves with more pronounced Group * Time interaction effects (see below and compare significance of between-group differences in respective Figs). Specifically, after a common rapid SVR drop in response to the transition from a standing to lying posture in both groups, there was a slow step-wise increase of the SVR curve level, as its main trend, in the FM group in contrast to its fast reshooting (rise back) with its further slow trend up to a higher level in the CT group during the clinostatic challenge. Also seen as a trend in the SVR curve in response to the transition from lying to standing in the FM group was a rapid, very strong SVR rise followed by a slow step-wise decrease of SVR curve to a level much higher than that detected in the lying posture. In contrast, in the CT group, a transient (rapid and short) orthostatic SVR drop to the SVR level at the onset of the lying posture was followed by a rapid rise much above the lying level in response to the transition from lying to standing with a fast modest decline to a relatively stable curve level as its new standing steady state without a further detectable trend. Interferences of fast SVR curve fluctuations superimposed on its trends and steady states were also different between the groups with faster fluctuations in the CT group compared to the FM group and with higher amplitudes during the clinostatic challenge in the CT group but during the ortho-static challenge in the FM group.

Time windows of significant between-group differences of the curves with simultaneous 95% CIs before and after adjustment for a covariate (height) were depicted in Fig 5B and 5C, respectively. The greatest number of significant between-group differences in the unadjusted SVR curves were obtained after lying down, which persisted for about half a minute during the posture (time window: 15–53 s) with higher SVR levels in the CT group compared to the FM group. This was also observed immediately after standing, and persisted through the entire

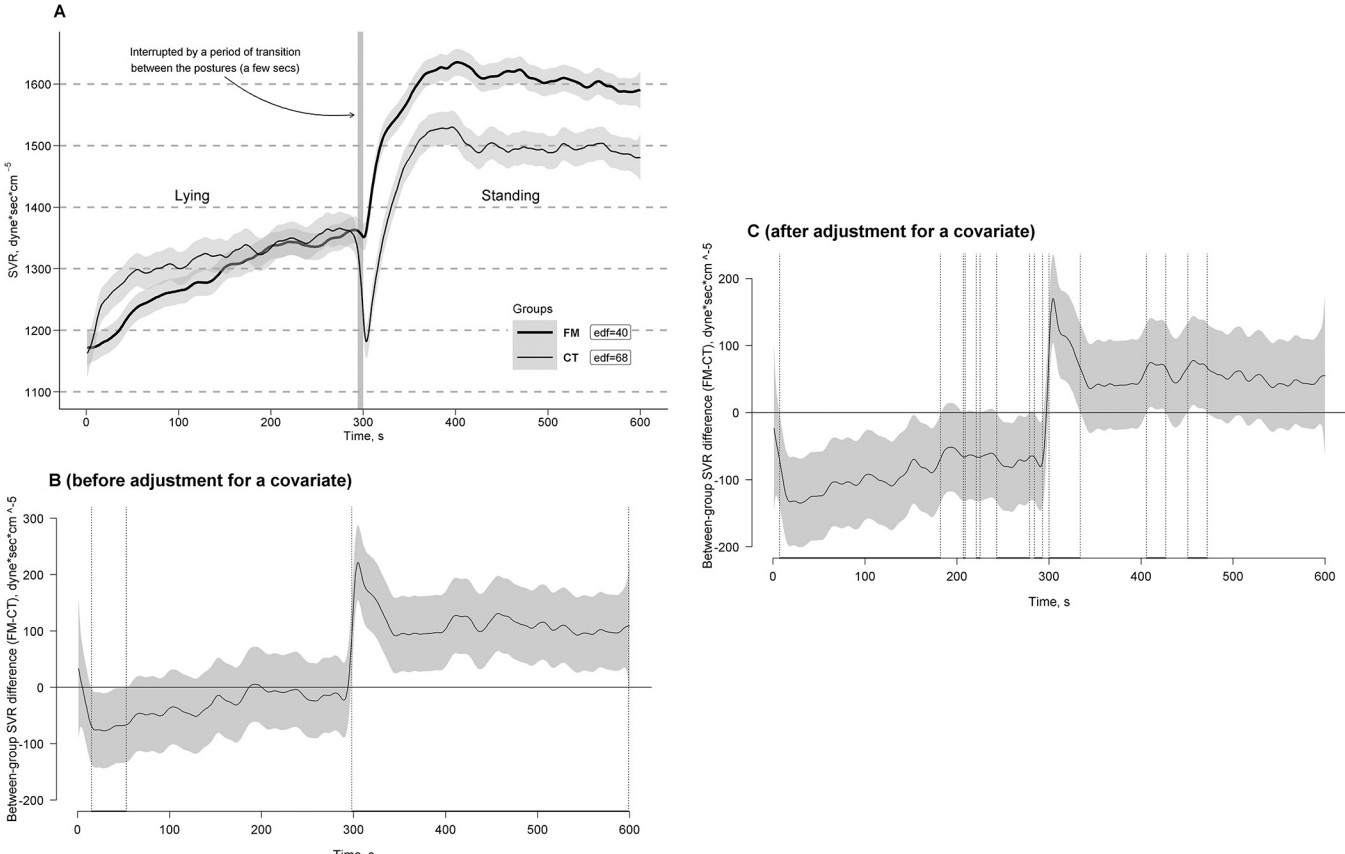

**Fig 5. Estimated time effects on systemic vascular resistance (SVR) curves during the clino- and ortho-static challenges.** The effects are averaged for the fibromyalgia (FM, bold line) and control (CT, thin line) groups, with approximate 95% confidence intervals (gray shading around the curves) and effective degrees of freedom or curve 'actual' turning points (edf) (A). Time windows showing significant between-group differences (FM-CT) are indicated by the position of SVR values with simultaneous 95% confidence intervals (gray shading around the curves) either above or below the zero line, without crossing it, and depicted as horizontal bold-line projections on the Time (*X*) axis, bounded by vertical dashed lines, corresponding to relevant points along the SVR difference curves, both before (B) and after (C) adjustment for a covariate (height).

posture (time window: 300–599 s) with higher SVR levels in the FM group compared to the CT group (Fig 5B). The greatest number of significant between-group differences in the adjusted SVR curves were obtained just after lying down with persistence for about 3 minutes during the posture (time window: 7–182 s) with a higher SVR curve level in the CT group compared to the FM group. This was also observed immediately after standing and persisted for about half a minute during the posture (time window: 300–334 s) with a higher SVR curve level in the FM group compared to the CT group. Due to periodic fluctuations of SVR curves, some additional shorter periods of significant between-group differences of SVR curves with the same direction of the grouping effects were found to be distributed over both postures (Fig 5C).

The correspondence between SVR and CO, as indicators of two components of BP regulation, was assessed. The results indicated a relative bias towards higher SVR dominance in BP control during the standing posture in the FM group (*CO/SVR* = 3.01[1.05] L*min*1000/dyne*sec*cm$^{-5}$) compared to the CT group (*CO/SVR* = 3.56[1.51] L*min*1000/dyne*sec*cm$^{-5}$), which served as the normal reference (*b*[SE] = -0.55 [0.28], *t* = -1.93, *p* = 0.057, bootstrap 95% CI = -1.14 to 0.00; see also Figs 3 and 5). Conversely, during lying, the correspondence between CO and SVR was similar in both groups (*CO/SVR* = 4.86[2.38] and 5.05

[3.67] L*min*1000/dyne*sec*cm$^{-5}$, respectively; $b$[SE] = -0.20 [0.68], $t$ = -0.29, $p$ = 0.775, bootstrap 95% CI = -1.63 to 1.12).

## Models for SBP

With 11.3% of explained variance in both GAMM and GAM, significant main Session, Group, and Posture linear effects were obtained for SBP with its higher mean during session 2 compared to session 1 ($b$[SE] = 0.52 [0.06], $t$ = 8.92, $p$ < 0.0001; here and further presented as in GAMM), in the CT group compared to the FM group ($b$[SE] = 7.63 [2.78], $t$ = 2.74, $p$ = 0.006), during lying compared to standing ($b$[SE] = -5.39 [0.40], $t$ = -13.42, $p$ < 0.0001), and in those with higher BMI ($b$[SE] = 1.01 [0.29], $t$ = 3.49, $p$ = 0.0005). Significant interaction Session * Group and Posture * Group linear effects were obtained for SBP from a similar level at session 1 to a higher rise in the CT group compared to the FM group by session 2 ($b$[SE] = 0.38 [0.09], $t$ = 4.48, $p$ < 0.0001) and with its deeper drop in the CT group compared to the FM group in response to the transition from a lying to standing posture reaching a close level in both groups ($b$[SE] = -5.14 [0.68], $t$ = -7.59, $p$ < 0.0001). A significant Grouping effect was obtained for SBP fluctuations during the clino-orthostatic challenge using $P$-spline adaptive smoothing models with different curve reconstructing criteria, defining "actual" or obtained numbers of curve turning points, i.e., edf values, equal to 33 for the FM group and 66 for the CT group ($F$ [29.42, 96827.89] = 11.30, $p$ < 0.0001) in GAMM and equal to 33 for the FM group and 63 for the CT group ($F$[33.35, 96834.43] = 0.36, $p$ < 0.0001) in GAM, i.e., with SBP fluctuation with about twice lower frequency in the FM group (i.e., about 3.3 cycles of SBP fluctuation per min during the challenges) compared to the CT group (i.e., about 6.3–6.6 cycles of SBP fluctuation per min).

Fig 6A shows the estimated time effects on SBP curves averaged per the FM and CT groups with approximate 95% CIs. As can be seen, a general trend of SBP curves in response to the clino- and ortho-static challenges (i.e., clino- and ortho-static SBP steady states) was similar in both groups with a slow decreasing trend of the SBP curves during the clinostatic challenge, a transient (rapid and short) SBP drop (deeper in the CT group) in response to the transition from a lying to standing posture with its fast reshooting (rise back), and absence of detectable changes in the trends of the SBP curves during the orthostatic challenge. However, an interference of fast SBP curve fluctuations superimposed on the trends were different between the groups. The fast SBP fluctuations were lower in amplitude and frequency in the FM group compared to the CT group making the SBP curve response flatter in the former group.

Time windows of significant between-group differences of the SBP curves with simultaneous 95% CIs before and after adjustment for a covariate (BMI) were depicted in Fig 6B and 6C, respectively. Significant between-group differences in unadjusted SBP curve responses were found immediately after the transition from both standing to lying and from lying to standing postures (respective time windows: 1–26 s and 300–335 s) with some other significant between-group differences mainly distributed randomly during the first minutes of the orthostatic posture (time windows: 364–392 s and 404–470 s) with deeper SBP curve drops in the CT group compared to the FM group (Fig 6B). A significant between-group difference in adjusted SBP curve responses was found immediately after the transition from a lying to standing posture (time window: 300–314 s) with an almost 4 times deeper SBP drop in the CT group compared to the FM group (Fig 6C).

## Models for DBP

With 12.5% of explained variance in both GAMM and GAM, significant main Session and Posture linear effects were obtained for DBP with its higher mean during session 1 compared

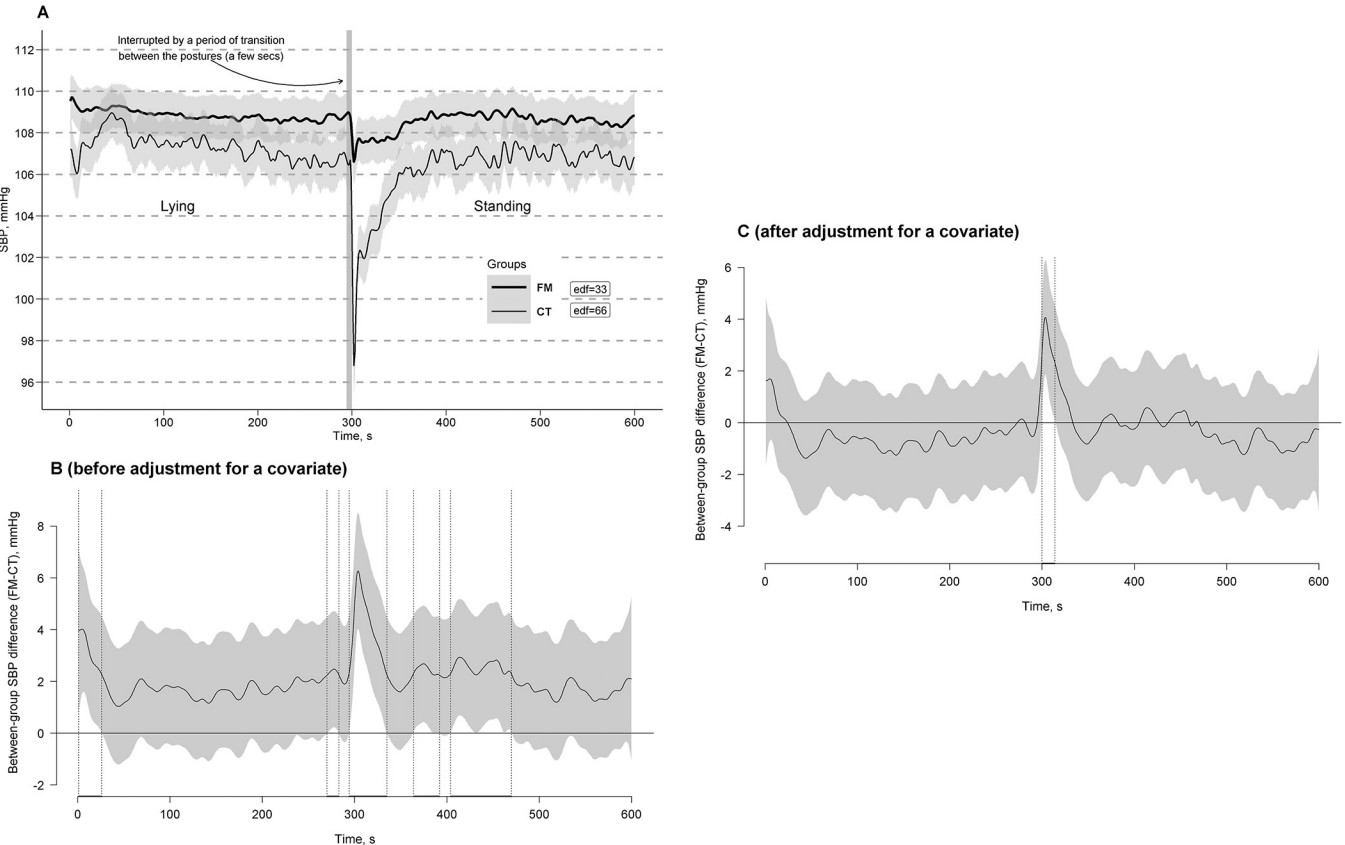

**Fig 6. Estimated time effects on systolic blood pressure (SBP) curves during the clino- and ortho-static challenges.** The effects are averaged for the fibromyalgia (FM, bold line) and control (CT, thin line) groups, with approximate 95% confidence intervals (gray shading around the curves) and effective degrees of freedom or curve 'actual' turning points (edf) (A). Time windows showing significant between-group differences (FM-CT) are indicated by the position of SBP values with simultaneous 95% confidence intervals (gray shading around the curves) either above or below the zero line, without crossing it, and depicted as horizontal bold-line projections on the Time (X) axis, bounded by vertical dashed lines, corresponding to relevant points along the SBP difference curves, both before (B) and after (C) adjustment for a covariate (body mass index).

to session 2 ($b$[SE] = -0.45 [0.04], $t$ = -10.45, $p < 0.0001$; here and further presented as in GAMM), during standing compared to lying ($b$[SE] = 3.49 [0.04], $t$ = 80.44, $p < 0.0001$), and in those with higher BMI ($b$[SE] = 0.74 [0.22], $t$ = 3.40, $p = 0.0007$). Significant interaction Session * Group and Posture * Group linear effects were obtained for DBP with its rise in the CT group and its drop in the FM group at session 2 from session 1 when levels were similar ($b$[SE] = 1.34 [0.06], $t$ = 21.39, $p < 0.0001$) and with its higher rise in the FM group compared to the CT group in response to the transition from a lying to standing posture ($b$[SE] = -2.95 [0.47], $t$ = -6.34, $p < 0.0001$). A significant Group effect was obtained for DBP fluctuations during the clino-orthostatic challenge using $P$-spline adaptive smoothing models with different curve reconstructing criteria, defining "actual" or obtained numbers of curve turning points, i.e., edf values, equal to 24 for the FM group and 41 for the CT group ($F$[34.50, 96872.57] = 26.71, $p < 0.0001$) in GAMM and equal to 24 for the FM group and 39 for the CT group ($F$[26.91, 96878.96] = 0.99, $p < 0.0001$) in GAM, i.e., with DBP fluctuation with about 1.7 times lower frequency in the FM group (i.e., about 2.4 cycles of DBP fluctuation per min during the challenges) compared to the CT group (i.e., about 3.9–4.1 cycles of DBP fluctuation per min).

Fig 7A shows the estimated time effects on DBP curves averaged per the FM and CT groups with approximate 95% CIs. As can be seen, general trends of DBP curves in response to the

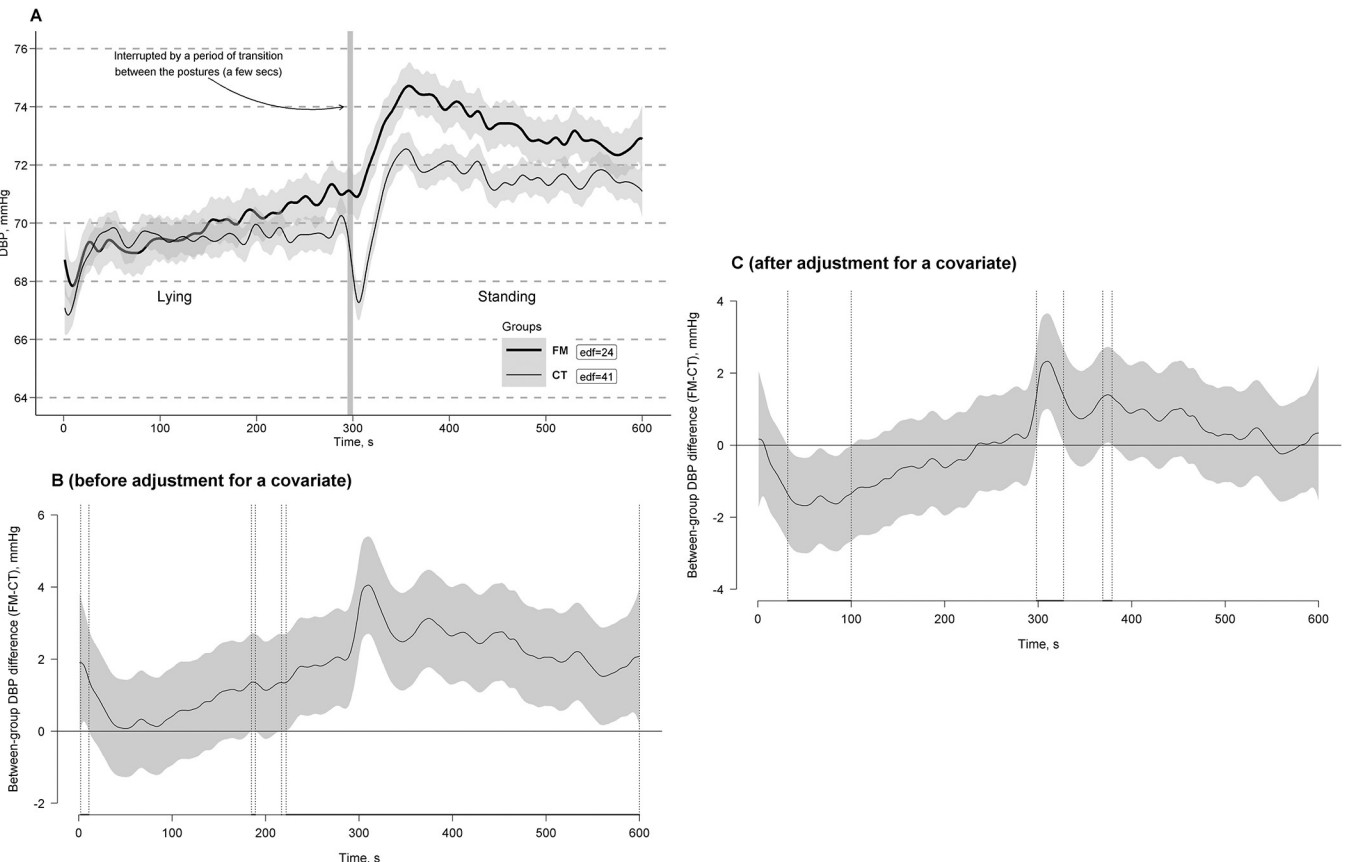

**Fig 7. Estimated time effects on diastolic blood pressure (DBP) curves during the clino- and ortho-static challenges.** The effects are averaged for the fibromyalgia (FM, bold line) and control (CT, thin line) groups, with approximate 95% confidence intervals (gray shading around the curves) and effective degrees of freedom or curve 'actual' turning points (edf) (A). Time windows showing significant between-group differences (FM-CT) are indicated by the position of DBP values with simultaneous 95% confidence intervals (gray shading around the curves) either above or below the zero line, without crossing it, and depicted as horizontal bold-line projections on the Time (*X*) axis, bounded by vertical dashed lines, corresponding to relevant points along the DBP difference curves, both before (B) and after (C) adjustment for a covariate (body mass index).

clino- and ortho-static challenges (i.e., clino- and ortho-static effects on the DBP steady states) were different in the groups with a slow step-wise increase of the DBP curve level as its trend in the FM group in contrast to a prompt rise of the DBP curve to a stable level without a detectable trend in the CT group during a clinostatic state after its common rapid drop in response to the transition from a standing to lying posture. A transition from lying to standing in the FM group was accompanied by a rapid, very strong DBP rise followed by a slow step-wise halfway decrease of the DBP curve as its main trend to a level higher than the one detected in the lying posture. In the CT group, a transition from lying to standing was accompanied by a transient (rapid and short) DBP drop with a later (i.e., delayed) rise much above the lying steady state level of the DBP curve with a fast half-way decline to a new stable level of the DBP curve as its standing steady state without any further detectable trends during the orthostatic challenge. Interference of fast DBP curve fluctuations superimposed on its trends and steady states were also different between the groups.

Time windows of significant between-group differences of the DBP curves with simultaneous 95% CIs before and after adjustment for a covariate (BMI) were depicted in Fig 7B and 7C, respectively. Significant between-group differences in unadjusted DBP curve responses were found immediately after the transition from a standing to lying posture (time windows:

1–10 s), during the last minutes of lying down (time window: 223–300), and through the whole orthostatic posture (time windows: 300–600 s) with a lower DBP curve level in the CT group compared to the FM group (Fig 7B). The greatest number of significant between-group differences in adjusted DBP curves were obtained shortly after lying down and persisting for about a minute during the posture (time window: 30–101 s) with a higher DBP curve level in the CT group compared to the FM group, and immediately after standing up and persisting for about a half minute during the posture (time window: 300–327 s) with a higher DBP curve level in the FM group compared to the CT group (Fig 7C). A further short time window (369–379 s) of significant differences in the orthostatic DBP curves between the groups could be related to a faster drop of the DBP curve level during standing in the CT group in contrast to a slower step-wise DBP curve level decrease in the FM group (see Fig 7A).

## Discussion

By combining non-linear (adaptive smoothing) models with conventional linear models, this study sought to investigate the association of chronic pain conditions with alterations in the parallel and sequential cooperation of hemodynamic processes, particularly focusing on certain cardiac (HR, CO, PEP) and vascular (SVR) parameters, as an integrated CV pain-o-metric pattern. Typically, the organism's compensatory or balancing reflexes integrate these processes into a unified cooperative mechanism to counteract gravity-induced shifts or translocations in blood volume during posture changes, akin to acute body re- and de-hydration events. These blood translocation events lead to either an increase or decrease in venous blood return to the heart, causing a corresponding rise or fall in cardiac preload and stroke volume. Stroke volume, being a key component of end-diastolic effective blood volume, is highly dependent on venous return and preload, making it a surrogate measure of both. These changes subsequently impact arterial BP and, ultimately, cerebral blood flow [12, 63–65] (see further discussion of these mechanisms in S Materials.II in S1 Materials).

### Stroke volume (SV) response: An initial challenging component of the CV pain-o-metric pattern

In line with previous research [14, 66–68], conventional linear models utilized in this study revealed the anticipated decrease in SV upon standing and the expected increase upon lying down. According to the linear model, these relative SV fluctuations between lying and standing positions were narrower in the healthy group compared to the FM group. This suggests that FM exacerbated the condition by amplifying the fluctuation of posture-related water shifts, with a more pronounced acute central rehydration after lying down and a more profound acute central dehydration after standing up, compared to the healthy group. However, employing a non-linear model with a smoothing data-driven approximation method revealed that the SV drop upon standing was not only more pronounced but also occurred more rapidly in the patient group compared to the healthy group, suggesting an initial challenging component in the CV pain-o-meter pattern. This accelerated and intensified SV drop in the patient group may be indicative of a swifter and more profound acute orthostatic dehydration compared to the preceding clinostatic hydration state, with the absolute magnitudes of dehydration and rehydration events contingent upon adjustments made for participant height and age.

In addition to identifying rapid aperiodic SV changes during clino-orthostatic transitions, this adaptive data-driven model-fitting method also detected prolonged slow SV trends as the primary steady states over time, along with their modulation by periodic fluctuations. While the slow trends were similar across groups, the modulation of these trends by periodic SV

fluctuations differed between them. A detailed discussion of these periodic fluctuations, along with the underlying mechanisms and the impact of chronic pain, is provided in S Materials. VIII in S1 Materials.

## Heart rate (HR) and cardiac output (CO) responses: Primary compensating components in CV pain-o-metric pattern

When a person stands up, the reduction in SV due to reduced venous return should trigger a cascade of reflex responses related to parasympathetic withdrawal and sympathetic nervous system activation [14, 66, 69]. This leads to an initial rapid HR increase characterized by a quick overshoot followed by a drop in the first few heartbeats, with HR then stabilizing at a steady state during standing. However, in healthy individuals, the initial HR overshoot is generally more pronounced, indicating a more effective cardiac baroreflex response that rapidly compensates for the orthostatic drop in SV, as evidenced by a brief transient increase in CO. In this group, the later HR stabilizes at a lower steady-state level, but still high enough to adequately support CO in response to gravity-induced blood volume translocation during standing. This effective orthostatic response may provide protection against cerebral hypoperfusion both at the initial standing phase and later while standing. In contrast, FM patients exhibit a higher steady-state HR with a less pronounced initial overshoot, indicating a diminished capacity to compensate for a more significant orthostatic drop in SV, as evidenced by an immediate orthostatic CO drop followed by stabilization at a lower steady state (for two possible explanations of this mechanism—either primary lower baroreflex sensitivity or its inhibition secondary to an unbalanced increase in cardiac afterload in the FM group—refer to S Materials.IV in S1 Materials). This reduced capacity to compensate for orthostatic central hypovolemia in FM patients could predispose them to cerebral hypoperfusion, potentially resulting in brain ischemia.

Animal studies have shown that following cerebral ischemia or insults, specific immuno-competent glial cells responsible for surveilling and maintaining homeostasis in the extracellular environment become hyper-activated, indicating that regular cerebral hypoperfusion events may lead to chronic central neuroinflammation [70, 71]. As a result, recurring episodes of cerebral hypoperfusion due to poorly compensated orthostatic blood redistributions may increase susceptibility to central sensitization, which involves heightened nociceptive signaling within the central nervous system. This process is considered a key factor in the pathogenesis of FM [72–76].

A reduced initial HR overshoot, along with a drop in CO, might suggest an impaired or decompensated initial phase in the CV pain-o-meter pattern, while a higher steady-state HR with a lower CO could indicate a similar impairment at a later stage, resembling postural orthostatic tachycardia syndrome (POTS), as seen in renin-dependent hypertension, chronic fatigue syndrome, and functional abdominal pain [8, 77, 78]. This sheds light on specific comorbidities and allows for the extrapolation of findings to other analogous chronic pain syndromes and forms of hypertension.

Along with identifying rapid, aperiodic HR changes during clino-orthostatic transitions and subsequent HR trends, this adaptive data-driven model-fitting method also detected modulation of these trends by periodic fluctuations at both low and high frequencies. While the low-frequency fluctuations were consistent across groups, the higher-frequency HR fluctuations varied between them. A detailed analysis of these periodic fluctuations, their underlying mechanisms, and the influence of chronic pain is provided in S Materials.VIII in S1 Materials.

## Pre-ejection period (PEP) and systemic vascular resistance (SVR) responses: Alternative compensating components of pain-o-metric pattern

The contrast between groups in HR and CO reactivities to standing, along with differences in HR and CO steady states during the challenge, was accompanied by distinct between-group variations in PEP and SVR responses to standing. These latter variations can serve as indicators of cardiac and vascular sympathetic reactivity, representing different patterns of $\beta$- and $\alpha$-adrenergic receptor stimulation that are mainly associated with changes in cardiac contractility and peripheral vascular activity [68, 79–83].

In this study, orthostatic PEP initially increased faster in the patient group during the early cardiac cycles compared to the healthy group, but subsequently reached a lower steady state level in the patients compared to the healthy participants. Consequently, patients experienced a rapid decrease in cardiac contractility upon standing, marked by a quicker negative inotropic response to offset the rapid onset of central hypovolemia, indicated by an accelerated orthostatic drop in SV. This suggests a properly functioning initial adaptation component within the CV pain-o-meter pattern. Unlike the initial rapid decrease in cardiac contractility that compensated for the accelerated drop in central blood volume upon standing, the FM group exhibited a relatively higher steady-state orthostatic cardiac contractility, as indicated by lower orthostatic PEP values, compared to the healthy group. This suggests that the reduction in cardiac contractility in the FM group was insufficient and remained disproportionately elevated relative to the greater central hypovolemia throughout the entire standing posture, as evidenced by a more pronounced drop in orthostatic SV compared to the healthy group. Conversely, healthy participants showed a slower negative inotropic response to standing, indicating a gradual reduction in cardiac contractility to offset the slower onset of central hypovolemia, reflected by a gradual orthostatic reduction in SV. They also maintained a lower orthostatic inotropic steady state, as indicated by higher orthostatic PEP values, suggesting a relatively greater reduction in cardiac contractility to more effectively manage central orthostatic hypovolemia, as reflected by their SV steady state. Hence, the initial orthostatic SV drops were appropriately compensated by the responsiveness of cardiac contractility in both groups, adhering to the Frank-Starling Law, which delineates the normal correlation between the length and tension of the myocardium: greater myocardial stretch during preload leads to stronger cardiac contraction. However, while the patient group displayed a relatively more pronounced reduction in SV during standing—suggesting a greater decrease in myocardial length extension due to reduced end-diastolic volume—this was not accompanied by a sufficiently effective decrease in cardiac contractility. This is evident in the less significant prolongation of the PEP associated with decreased SV in the patient group (7.19 mL*sec) compared to healthy participants (8.03 mL*sec) (see Figs 1 and 4). Conversely, while lying down, elevated end-diastolic volumes in both groups corresponded similarly to increased cardiac contractility, as evidenced by the similar products of PEP and SV values between the groups (9.13 and 9.21 mL*sec, respectively). This suggests a potential dysfunction in the Frank-Starling mechanism governing the length-contraction or SV-PEP relationship in cardiac muscles during standing in patients with FM, indicating an impaired or decompensated later component in the CV pain-o-meter pattern.

Hence, it could be inferred that, for a given SV, which serves as a proxy measure associated with preload or left ventricular end-diastolic pressure, the left ventricle appeared to exert greater effort during standing in FM patients compared to what is expected according to the healthy reference group. This suggests a violation of the Frank-Starling Law in these individuals during this challenge posed by gravity. The elevated hemodynamic stress on the heart during standing in the patient group did not appear to be linked to disruptions in inotropy, as it

effectively balanced the high SV levels during lying and the SV reduction dynamics shortly after standing in both groups. Instead, this additional cardiac workload likely stemmed from increased vascular tone during standing, possibly due to higher vascular $\alpha$-adrenoceptor activity, necessitating greater cardiac effort to overcome increased cardiac afterload [84].

Indeed, alongside changes in cardiac inotropy or contractility, as assessed by PEP, vascular tone, as assessed by SVR, showed a contrasting pattern between the healthy and patient groups after standing. While the healthy group experienced a rapid drop in SVR followed by a gradual rise to a steady state, the patient group exhibited an immediate rise in SVR to a higher level of the steady state. This indicates a biphasic physiological response in the healthy group versus a monophasic response in the patient group, implying initial and later compensating or adaptation components in the CV pain-o-meter pattern. Therefore, these findings support the idea, that in the latter group, the disproportionately heightened cardiac contractility observed during standing could be linked to the additional cardiac workload required to overcome the extra systemic vasoconstriction (i.e., higher afterload). This alternative CV mechanism in the patient group compensates for a more substantial drop in orthostatic CO, which serves as the primary hemodynamic mechanism in the healthy group, with both mechanisms balancing to maintain orthostatic BP within the homeostatic range.

These findings of heightened systemic vasoconstriction during standing in patients with fibromyalgia align with the suggested role of neuroinflammation in its pathophysiology, contributing to central sensitization as its primary characteristic. In fibromyalgia, neuroinflammation linked to central sensitization, coupled with systemic inflammation associated with peripheral sensitization, may primarily result from the interplay of regular orthostatic cerebral hypoperfusion events with recurrent systemic vasoconstriction. This vasoconstriction serves as the principal alternative compensatory mechanism in BP regulation, albeit with sustained tissue hypoperfusion or functional microvessel rarefaction as a side effect [7]. Hence, these poorly regulated systemic (central and peripheral) hemodynamic responses to regular orthostatic challenges could potentially act as triggers in the cascade from central and peripheral inflammation to central and peripheral sensitization, ultimately contributing to the development of fibromyalgia as a heterogeneous chronic pain syndrome or worsening the severity of existing fibromyalgia symptoms. In addition to fibromyalgia, chronic neuroinflammation has also emerged as a significant factor in the development of other chronic pain syndromes, including neuropathic and chronic low back pain and may present new avenues for their treatment [71, 76, 85, 86].

A delayed negative inotropic response (i.e., a slower decline in cardiac contractility) to standing in healthy participants, indicated by a delayed initial orthostatic prolongation of PEP, is potentially related to an immediate increase in venous blood return and central blood volume after standing up. The later effect is driven by a complex mechanism that typically counteracts orthostatic gravity-induced blood redistribution to the lower body. This includes immediate transient peripheral vasodilation—possibly associated with pulmonary baroreceptors and peripheral ATP-related mechanisms—along with a rapid shift of unstressed blood reserves from splanchnic and muscular vessels due to abdominal and calf muscle contractions, effectively acting as a skeletal muscle 'pump' during the initial cardiac cycles [50, 64, 69, 79–82, 87, 88] (see an alternative explanation and discussion of the mechanism controversies in S Materials.III in S1 Materials).

Since most FM patients in the present study appeared to lack blood reserves due to total hypovolemia [7], they missed this short-lived compensatory mechanism involving cardiac contractility and vasodilation, preventing them from maintaining effective central blood volume at this initial stage. Although there were also differences in orthostatic vasoconstriction between the groups during the steady state, these balanced out the orthostatic CO differences,

resulting in similar SBP levels in both groups during this phase. This suggests that the 'servo-mechanism' regulating vascular capacitance relative to circulatory blood volume, which together support BP within a homeostatic range, functioned properly in both groups [6, 89–91]. However, in FM patients, this compensating 'servo-control' mechanism during the ortho-static steady state appeared to rely more heavily on vasoconstriction, with a CO to SVR ratio of 3.01 L*min*1000/dyne*sec*cm$^{-5}$, compared to 3.56 L*min*1000/dyne*sec*cm$^{-5}$ in the healthy group. This is despite the CO and SVR balances being closer in both groups during the clino-static steady state, with the ratio values of 4.86 and 5.05 L*min*1000/dyne*sec*cm$^{-5}$, respectively. These findings confirmed that the FM cohort experienced additional cardiac workload due to increased cardiac afterload while standing, as evidenced by the product of SV and PEP, which deviated from the expected value as per the Frank-Starling law (see above).

In addition to identifying rapid, aperiodic changes in PEP and SVR during clino-orthostatic transitions and their subsequent trends, the adaptive data-driven model-fitting method also detected modulation of these trends by periodic fluctuations, primarily at low frequencies around 0.1 Hz. While the PEP fluctuations were consistent across groups, the SVR fluctuations differed, occurring at a lower rate in patients. A detailed analysis of these periodic fluctuations, their underlying mechanisms, and the impact of chronic pain is provided in S Materials.VIII in S1 Materials.

## Systolic blood pressure (SBP) and diastolic blood pressure (DBP) responses: Homeostatic components in pain-o-metric pattern

Typically, BP levels and their fluctuations are perceived as 'servo-balanced' homeostatic outcomes, wherein changes in CO due to various challenges are compensated by adjustments in SVR, ensuring a dynamic equilibrium between cardiac preload and afterload mechanisms [12]. Nonetheless, in the healthy group of the present study, a rapid and pronounced initial drop in both SBP and DBP upon standing was observed, which attributed to an initial transient vasodilation, a phenomenon often referred to as 'initial orthostatic hypotension' [68] (additional insights into the functionality of this phenomenon beyond the 'servo-balanced' mechanism are offered below and in the S Materials.III and S Materials.IV in S1 Materials). This phenomenon is generally viewed as normal and short-lived, with a temporary rise in CO resulting from the reversal of the 'native' servo-mechanism to boost central hemodynamics through an increase in central blood volume, as seen in this study. Subsequently, BP recovery happens as systemic vascular tone rises, compensating for the decline in CO, which indicates the reestablishment of hemodynamic control via the 'native' servo-mechanism. In contrast, the patient group exhibited other responses as initial components of the pain-o-metric pattern, marked by a very weak initial decline in SBP coupled with an immediate rise in DBP, attributed to the absence of the initial vasodilation. These initially contrasting BP responses were eventually followed by a recovery similar to that in the healthy group, though with a delayed and less stable steady state for DBP. The latter finding signifies another subsequent or later component of the pain-o-metric pattern, attributed to an overactivated and unstable systemic vascular tone, accompanied by increased cardiac workload due to elevated heart rate and contractility in the group of FM patients.

In addition to identifying rapid, aperiodic changes in SBP and DBP during clino-orthostatic transitions and their subsequent trends, the adaptive data-driven model-fitting method also detected modulation of these trends by periodic fluctuations, primarily at low frequencies. The rates of SBP and DBP fluctuations differed between groups, with patients exhibiting a rate two times lower for both. Additionally, in both groups, SBP fluctuations occurred at a higher rate than DBP fluctuations. A detailed analysis of these periodic fluctuations, their underlying mechanisms, and the impact of chronic pain is provided in S Materials.VIII in S1 Materials.

In contrast to other hemodynamic and CV measures, the differences in SBP and DBP activity between the FM and healthy groups during the clino-orthostatic challenge diminished after adjusting for BMI as a covariate (Figs 6D and 7D). This is likely because BMI, a key covariate for these metrics in both the GAMM and GAM models, differed significantly between the groups (Table 1). As a probable mediating factor in the causal mechanisms linking chronic pain and BP [1], BMI likely contributed to these between-group differences. In contrast, covariates such as age and height, though more relevant to other measures, had a lower and statistically insignificant impact on between-group differences, despite their significant linear effects in the models. These covariates primarily influenced overall between-subject variability, contributing less to between-group differences.

## Mechanisms underlying components of the pain-o-metric pattern

While the impaired and adaptive initial hemodynamic and CV responses of FM patients to the active orthostatic posture in this study closely resembled those observed in response to the passive orthostatic position in previous studies on healthy populations, this consistency did not extend to the impaired and adaptive late hemodynamic and CV steady states in FM patients [50]. The former consistency can be explained by the absence or reduction in FM patients of the effects of CV mechanisms usually responsible for early responses to active orthostatic challenges, similar to those typically absent or reduced during passive orthostatic challenges like head-up tilting. These typically active orthostatic mechanisms involve rapid and adequate positive chronotropic cardiac reactivity, along with only a mild initial decrease in cardiac contractility to balance the active venous return resulting from central blood volume shifts from the splanchnic and other lower-body regions, which is facilitated by intra-abdominal pressure and skeletal muscle contractions. This process is supported by cardiopulmonary baroreflexes and the release of local autoregulation agents responsible for rapid transient systemic vasodilation (see discussion of the mechanism controversies in S Materials.III and S Materials.IV in S1 Materials).

In contrast, mechanisms regulating later hemodynamic and CV responses to active standing in FM patients, resulting in overactivated DBP, HR, and vasoconstriction coupled with decreased SV and CO, seemed to differ from the mechanisms typically affected in the later CV states during passive head-up tilting, which resulted in decreased DBP and HR, coupled with SV, CO, and vasoconstriction comparable to those observed during the active standing posture [50, 69, 88]. This supports the notion of the predominant role of the adaptive hemodynamic 'servo-control' mechanism in FM patients, favoring the overactivation of cardiac afterload mechanisms to compensate for the underactivity of cardiac preload mechanisms during the late phase of orthostatic challenge. It signifies the main component of the pain-o-metric pattern, attributed to the total body water deficit or hypohydration state, which determines deficits in total stressed (effective) and unstressed blood volume reserves [7]. Although these key elements of the pain-o-metric pattern were identified in hemodynamic and CV responses to orthostatic challenges, clinostatic hemodynamic and CV responses also exhibited pain-o-metric characteristics linked to the same total blood volume deficit, supporting this mechanism as a potential root cause (see more details in the S Materials.V in S1 Materials).

Thus, in contrast to the healthy group, FM patients seem more prone to cardiac and cerebral hypoperfusion events during the early and later stages of orthostatic challenge due to two main factors. First, the physiological protection against the initial significant orthostatic SV drops for stabilizing central body blood volume is compromised in FM patients by a combination of sluggish and insufficient cardiac positive chronotropic reactivity and a steep, pronounced decrease in cardiac contractility, stemming from deficits in the mechanisms

responsible for active cardiac venous return from the lower-body regions. Second, the physiological mechanism to stabilize BP by compensating for decreased SV during the later stage of an orthostatic challenge is compromised due to an imbalance in these systems. This results in increased cardiac afterload because of a diminished role of preload due to lower blood volume reserves, subsequently overburdening the heart with higher inotropic and chronotropic activity. Thus, under normal conditions, the two successive initial and later stages of hemodynamic regulation in response to clino-orthostatic challenges work in tandem to stabilize central blood volume and maintain consistent BP. This ensures a continuous and efficient flow of blood to the brain and peripheral tissues and provides them with oxygen and nutrients to meet their metabolic demands. In FM patients, alternative mechanisms of hemodynamic regulation in response to clino-orthostatic challenges maintain consistent BP without stabilizing central blood volume. This leads to continuous but inefficient blood flow to the brain and peripheral tissues, which results in an oxygen and nutrient deficit when facing these challenges. These alternative mechanisms could increase the likelihood of recurrent orthostatic syncope events in these individuals, raising the risk of cardiac and brain ischemia and potentially triggering peripheral and central inflammation. This, in turn, may contribute to the development of various chronic pain syndromes, including fibromyalgia. This is particularly likely to happen in individuals with limited capacity for vasoconstriction during states of total body hypovolemia or hypohydration [69, 80, 92].

In this context, interventions designed to increase total body water capacity with a focus on enhancing blood volume reserves—as part of physiological resilience reserves—are expected to offer considerable benefits for individuals with overstressed vasoconstriction mechanisms nearing their limits for cardiovascular adaptation to reduced blood volume. This enhancement of the physiological resilience reserves is expected to improve orthostatic tolerance—a critical indicator of overall health [1, 16, 17, 92]. S Materials.VII in S1 Materials and earlier publications [1, 7] offer detailed insights into various procedures and interventions designed to boost resiliency against chronic pain syndromes through the hemodynamic and CV mechanisms outlined in this study. This highlights their potential role in preventive and therapeutic approaches for chronic pain. Findings from this study, along with insights from related previous research, also suggest additional strategies for addressing orthostatic hypoperfusion by strengthening the mechanisms that compensate for central and peripheral blood redistribution failures. These strategies could involve training or conditioning procedures to bolster the observed between-session effects that contribute to improvements in specific transient and long-lasting hemodynamic and CV responses to gravitational stress (see S Materials.I in S1 Materials).

## Advantages and limitations of the present study

The consistency between the findings of this study—achieved using data-driven non-linear models to analyze hemodynamic and CV processes across the entire challenge timeframe as parts of a unified body mechanism—and those from earlier studies, which employed separate theoretically driven linear models for the same processes at different timeframes, along with the between-session congruence observed in this study, underscores the reliability and robustness of the results obtained in the FM group through this advanced analytical approach (see also S Materials.VI and S Materials.I in S1 Materials). However, despite the consistency with previous studies, this study also revealed that, compared to non-linear models, linear models were unable to detect between-group differences in speed, velocity, or rate of CV responses (e.g., faster drop/rise versus slower decrease/increase, or steeper versus flatter slope), as well as the time duration of CV responses, periodic and aperiodic fluctuations, non-linear trends, and delays.

The adaptive model-fitting approach used in this study can indicate not only fast aperiodic CV reactions, like initial clino-orthostatic HR and BP responses, but also periodic fluctuations in certain hemodynamic processes. These processes involve interactions between the periodic unloading/loading of low-pressure baroreceptors (volume mechano-receptors) in the large systemic veins, heart, and pulmonary vessels, and high-pressure baroreceptors in the carotid and aortic arteries, as well as respiratory sinus arrhythmia, all influencing cardiac and vascular activities to varying degrees. This interplay results in chronotropic modulation of these activities across different frequencies. Notably, the study found that FM affects the regulatory mechanisms of these fluctuations, as evidenced by group differences in the effective degrees of freedom (edf)—representing the 'actual' turning points of hemodynamic and CV curves—obtained using the data-driven analytical technique. Additional insights into the application of this technique in this context can be found in S Materials.VIII in S1 Materials, which adds components related to periodic fluctuations in hemodynamic and CV processes to the specified pain-o-metric profile. Although these fluctuating processes were not extensively explored in this study, it is worth highlighting that the analytical approach employed here offers a promising alternative for their analysis. Compared to conventional methods like autoregressive models or the Fourier Transform, this approach has the potential to offer more insightful and interesting perspectives in this context [9, 22].

The results of this study were derived from FM patients as a group representative of the population with this syndrome. However, variations in the chronicity and severity of the condition could lead to different orthostatic CV and hemodynamic reactions and states [10]. This issue is being addressed in a follow-up study utilizing the same dataset, incorporating relevant measures of disease chronicity and severity. Another limitation is that this study involved relatively young female participants, both in the healthy group and among those with FM, a chronic pain syndrome. FM, as a pathological condition, is often misdiagnosed and underreported in general, with underdiagnosis in men in particular due to gender-biased and physiologically unjustified diagnostic tools for assessing FM symptoms, such as clinical evaluation of 'tender points' [93, 94]. In this context, the results of this study suggest the need to reconsider diagnostic criteria, as the syndrome may have a more severe pathophysiological impact in different sexes and age groups without manifesting clinical symptoms, leading to underreporting and potentially worse somatic outcomes. Some prior studies have found variations in orthostatic cardiac afterload (vasoconstrictor) and preload reserves based on gender and age. For example, healthy women were reported to have less orthostatic vasoconstrictor reserve than healthy men [95], while older individuals (69 ± 3 years) exhibited insufficiency in both cardiac preload (blood volume and contractility) and afterload (vascular pressor) mechanisms for effectively restoring orthostatic BP [96]. However, other studies have indicated that the orthostatic vascular pressor mechanism can remain unchanged or even increase with age, particularly in women [97, 98]. This variability indicates that systemic vasoconstriction—a crucial mechanism regulating BP stability in response to active standing, especially in relatively young Caucasian female FM patients, as observed in the present study—requires further investigation to establish its consistency as a CV pain-o-metric marker across different populations with chronic pain. Additionally, since pathological BP elevation has been linked to combined blood volume and vasoconstriction regulation mechanisms (e.g., through modulation of renin release and $\alpha_1$-adrenoreceptor sensitivity) in African Americans, exploring blood volume-related CV mechanisms as potential pain-o-metric markers in various ethnic groups is also warranted [99]. Thus, a notable limitation of this study was also the lack of significant cultural diversity of both the patient and healthy participant groups.

Some concerns may arise regarding potential confounding effects related to certain lifestyle factors such as cigarette smoking, regular alcohol consumption, and daily physical activity

habits on the observed associations between hemodynamic processes, CV mechanisms, and the chronic pain syndrome. However, previous research suggests that these lifestyle factors do not significantly alter the fundamental relationships between mechanisms governing CV processes and pain mechanisms [8, 16, 100–102]. These lifestyle factors exhibit complex effects; for instance, physical activity can either alleviate or worsen pain severity through autonomic or metabolic mechanisms, depending on the types of exercises undertaken by individuals with varying autonomic or metabolic phenotypes, thus impacting their pain sensitivity [100]. In the present study, many of these factors within the healthy group closely mirrored those in the FM group, aiming to reduce potential biases in the main results that might arise from confounding effects (see Table 1).

Hemodynamic measurements during the brief 2 to 3 second period of rising from a supine position were omitted in the present study, as they are prone to significant measurement error caused by finger cuff displacement during the balance adjustments needed to stand. This raises the possibility that some rapid CV changes, potentially triggered by the muscle-heart reflex, were missed during this time, which could be relevant to the primary hypothesis, experimental models, and underlying mechanisms in the findings. However, this experimental procedure did capture the expected transient HR peak and the corresponding transient rise in CO following standing up, typically linked to arterial baroreceptor inhibition. This observation aligns with previous studies, suggesting that hemodynamic changes observed immediately after standing can be considered sufficiently representative, as the effects of a few seconds during the transition to standing have a lasting impact [103].

Additionally, the present study could not definitively determine why BP regulation in FM patients seemed to rely on cardiac afterload rather than mechanisms that increase cardiac preload through physiological and behavioral means that promote central and total body hydration. While a decreased tendency to drink water might be influenced by a higher threshold for thirst sensation, increased water loss could be driven by elevated body temperature, which may suppress the body's ability to conserve water and salt. This issue may be closely linked to the recent public call for research on the contribution of global climate change to the global burden of mental health problems, in addition to its effects on physical health (e.g., those related to hypohydration, such as hypotension and certain subtypes of hypertension [7, 65]). Investigating the pathophysiological mechanisms behind this causal link, which remains poorly understood and under-researched, is crucial for guiding policymakers and stakeholders in developing interdisciplinary solutions to enhance heat resilience globally, especially in populations living in hot climates, like those studied here [104]. The indicated shift from conventional to alternative physiological mechanisms could also result from either impaired centrally regulated vasodilation or a deficit in peripheral vascular tone autoregulation, potentially leading to functional microvessel rarefaction. Both of these mechanisms could impact the pressure natriuresis-diuresis process, requiring higher BP or hypertensive states to balance sodium intake and output, as suggested by earlier studies [105, 106]. Recent studies have begun to address this knowledge gap, suggesting that long-term stress-related cortisol release may play a key role in shifting BP regulation from cardiac preload to afterload mechanisms in these patients [23]. Follow-up studies are further investigating this by using additional datasets that include measures to examine the interaction between thirst threshold, water-electrolyte balance, metabolic activity, and body temperature regulation across periods of natural weather variation, as well as lifestyle habits related to fluid intake. This will enable healthcare providers to offer more precise recommendations for personalized interventions aimed at restoring balance in BP regulation through the preload mechanism. Targeted approaches, such as vagus nerve stimulation to enhance centrally regulated vasodilation or thermal therapies (cold, hot, or contrast bathing) to modulate peripheral vascular tone, are also planned for exploration,

alongside more general methods like physical reconditioning. S Materials.VII in S1 Materials discusses, as additional perspectives, various potential techniques for restoring balance between central and peripheral hemodynamic mechanisms in cases of body water balance regulation challenges, particularly for proper blood volume redistribution in response to clino-orthostatic changes. These approaches are proposed to reduce the risk of developing chronic pain by leveraging the understanding of the unified complex hemodynamic mechanism and its potential effects on pain chronification mechanisms presented and discussed in the current study.

## Perspectives

Because treatment approaches for chronic pain syndromes, particularly FM, frequently rely on subjective symptoms, objective 'pain-o-meter' instruments have been proposed to improve treatment effectiveness. The absence of typical short-lived initial CV responses to rapid standing—such as transient vasodilation, a quick rise in CO, and a transitory HR increase—could be used as a pain-o-metric instrument to objectively measure chronic pain. These transient orthostatic responses can be easily detected using mobile devices that integrate CV measures with signals from accelerometers and gyroscopes [107, 108]. This approach could provide a way to monitor chronic pain and evaluate the effectiveness of its management in everyday settings. Additionally, understanding of physiological mechanisms responsible for the observed hemodynamic and CV changes in patients with chronic pain syndromes that could be associated with maladaptive effects from rapid pulmonary and cardiac baroreflexes, from prompt and delayed central autonomic and local autoregulatory mechanisms as well as rapid abdominal and low limb muscle contractions, may help in the development of new technologies for pain syndrome managements and its prevention (see more details in S Materials.VII in S1 Materials).

The findings also suggest that in cases of comorbid pain and hypertension, where a 'blood pressure-related hypoalgesia' phenomenon is observed (see the introduction), antihypertensive medications with diuretic effects (such as thiazide diuretics, mineralocorticoid receptor blockers, calcium channel blockers, and $\alpha_1$-adrenergic blockers) should be avoided, as they may increase central hypovolemia. Additionally, dietary recommendations with sodium restrictions can have a similar negative impact. Furthermore, medications with vasodilatory effects (including angiotensin-converting enzyme inhibitors, angiotensin II receptor blockers, direct renin inhibitors, $\beta$-adrenergic blockers, or centrally acting $\alpha$-agonists) might disrupt the vasoconstriction 'adaptation to pain' mechanism, potentially worsening pain severity and chronicity unless combined with strategies to restore cardiac preload, such as improving venous blood return, which serves as a physiological resilience mechanism. Beyond the clinical significance of these findings, an important healthcare implication is the ability to identify at-risk populations and recommend appropriate protective or preventive actions. For example, conditions or situations that lead to blood volume contraction (like intense sweating, diuresis, or fluid deposition in the skin and muscles) with a tendency toward cardiac afterload regulation of BP should be addressed with simple procedures to increase stressed or effective blood volume, such as enhanced salt and water intake.

The primary aim of this research was to examine the significance of variations in hemodynamic and CV processes as a unified mechanism, with a focus on their collective alterations in relation to chronic pain and their diagnostic potential compared to controls. However, specific components of this mechanism, coupled within submechanisms, may have greater value for therapeutic monitoring, while others may serve as prognostic markers. For instance, the rapid, transient coupling of HR and BP responses to standing—an early key indicator of

hemodynamic regulation—could serve as a 'pain-o-metric' tool for tracking therapeutic progress and effectiveness in chronic pain management. This coupling could be assessed both in clinical settings and through ambulatory or wearable monitoring devices. Additionally, the dynamics and trends of CO and SVR in response to clinostatic-to-orthostatic transitions—reflecting the balance between preload and afterload mechanisms related to central blood volume reserves in stabilizing arterial pressure and blood flow—could serve as a prognostic tool for assessing the broader negative impact of chronic pain on mental and physical health by compromising adequate blood supply to vital organs such as the brain, kidneys, heart, endocrine glands, and muscles. However, most of these mechanisms can be monitored and assessed with the use of appropriate digital and mobile health technologies.

## Conclusions

This work builds on previous research into the relationship between chronic pain and CV processes. However, unlike earlier studies, it is the first to examine these processes as part of a unified homeostatic mechanism for regulating BP and blood flow, accounting for phase shifts between individual hemodynamic and CV processes, their amplitude and frequency modulations, and rates of change over short (seconds) and long (minutes) periods in response to repeated clino-orthostatic, gravity-related challenges in both normal and pathological conditions. To address this complex task, the study utilizes an adaptive, data-driven method to automatically identify oscillations and states characteristic of both normal and pathological conditions, assessing the significance of differences specific to patients with chronic pain.

In summary, the non-linear models from this study revealed a close interaction between various hemodynamic and CV mechanisms across different timeframes and phases in response to recurrent clino-orthostatic challenges, illustrating their collective role in the context of chronic pain. The findings suggest that FM patients tend to depend more on cardiac afterload to regulate BP, employing an emergency 'servo-control' mechanism that reduces vascular capacity to counteract the blood volume deficit commonly observed in this population. This shift places greater strain on the heart, particularly in terms of contractility and heart rate, to manage the increased afterload and effective blood volume deficit, respectively. Although this alternative CV response to clino-orthostatic challenges can maintain consistent BP, it fails to stabilize venous blood return and central blood volume, leading to continuous yet inefficient blood flow to the brain and peripheral tissues. Findings of group differences in periodic amplitude variations, as long-term hemodynamic and CV fluctuations modulated trends at different frequencies, further reinforced the main results by linking these variations to the same mechanisms that explain both early short-lived aperiodic reactions and the later sustained trends in these processes. This can potentially lead to chronic peripheral tissue and cerebrovascular hypoperfusion, ischemia, or other insults, resulting in oxygen and nutrient deficits, which in turn could cause chronic brain and systemic inflammation, ultimately contributing to central and peripheral pain sensitization—the key mechanism underlying fibromyalgia and its severity.

The study also found that, in some cases, these alternative afterload-based mechanisms for BP control could be rebalanced toward more preload-focused mechanisms, impacting both BP and blood volume regulation, by undergoing recurrent clino-orthostatic challenges. This suggests a possible delay in the reconditioning process of mobilizing blood volume reserves. Further research is needed to determine whether these variations in delayed CV reconditioning effects, associated with CV health, are also linked to variations in chronic pain severity symptoms.

## Supporting information

**S1 Materials.**
(PDF)

## Acknowledgments

We would like to thank the Fibromyalgia Association of Jaén (Spain) and all participants for their contribution to the study.

## Author Contributions

**Conceptualization:** Dmitry M. Davydov.

**Data curation:** Dmitry M. Davydov, Carmen M. Galvez-Sánchez, Gustavo A. Reyes del Paso.

**Formal analysis:** Dmitry M. Davydov.

**Funding acquisition:** Gustavo A. Reyes del Paso.

**Investigation:** Dmitry M. Davydov, Carmen M. Galvez-Sánchez, Gustavo A. Reyes del Paso.

**Methodology:** Dmitry M. Davydov, Gustavo A. Reyes del Paso.

**Resources:** Gustavo A. Reyes del Paso.

**Supervision:** Gustavo A. Reyes del Paso.

**Validation:** Dmitry M. Davydov.

**Visualization:** Dmitry M. Davydov.

**Writing – original draft:** Dmitry M. Davydov.

**Writing – review & editing:** Dmitry M. Davydov, Gustavo A. Reyes del Paso.

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
