## [Decision Letter · Decision Letter 0]

11 Sep 2024

PONE-D-24-27267Hemodynamics in chronic pain: A pathway to multi-modal health risksPLOS ONE

Dear Dr. Davydov,

Thank you for submitting your manuscript to PLOS ONE. After careful consideration, we feel that it has merit but does not fully meet PLOS ONE’s publication criteria as it currently stands. Therefore, we invite you to submit a revised version of the manuscript that addresses the points raised during the review process.

We look forward to receiving your revised manuscript.

Kind regards,

Niema M. Pahlevan, PhD

Academic Editor

PLOS ONE

“Gustavo A. Reyes del Paso

the Spanish Ministry of Science and Innovation, co-financed by European Regional Development Fund [PID2022-139731OB-I00]”

4. In this instance it seems there may be acceptable restrictions in place that prevent the public sharing of your minimal data. However, in line with our goal of ensuring long-term data availability to all interested researchers, PLOS’ Data Policy states that authors cannot be the sole named individuals responsible for ensuring data access (http://journals.plos.org/plosone/s/data-availability#loc-acceptable-data-sharing-methods).

Reviewers' comments:

Reviewer's Responses to Questions

**Comments to the Author**

1. Is the manuscript technically sound, and do the data support the conclusions?

Reviewer #1: Yes

Reviewer #2: Yes

Reviewer #3: Yes

2. Has the statistical analysis been performed appropriately and rigorously? 

Reviewer #1: Yes

Reviewer #2: Yes

Reviewer #3: Yes

3. Have the authors made all data underlying the findings in their manuscript fully available?

Reviewer #1: Yes

Reviewer #2: No

Reviewer #3: No

4. Is the manuscript presented in an intelligible fashion and written in standard English?

Reviewer #1: Yes

Reviewer #2: Yes

Reviewer #3: Yes

5. Review Comments to the Author

Reviewer #1: This research article investigates hemodynamic and cardiovascular (CV) mechanisms in response to clino-orthostatic challenges in a population of fibromyalgia patients and compared it to a group of healthy controls. The novelty of the study stems in the implementation of non-linear models to capture the interactions between the distinct yet interconnected hemodynamic processes in the CV system, which cannot be captured using linear models. The authors seek to advance the CV ‘pain-o-meter’ hypothesis that aims to objectively assess chronic pain conditions based on measured CV mechanics. Overall, the paper is well written.

The reviewer would like the following points to be addressed:

- In the introduction, the authors claim that linear statistical methods were a limitation for analyzing the hemodynamic and CV reactions. As the methodological novelty of this paper originates from implementing non-linear models, it would be beneficial to expand on the current contributions of linear statistical models and where these have resulted in limited or erroneous understanding of hemodynamic processes.

- The statistical test performed and reported table 1 is not mentioned in the text. Please add a description of what has been performed and reported.

- The results of the administered questionnaires are presented in table 1, yet the description of these tests is described later in the manuscript. I suggest the authors consider reorganizing the flow.

- In the calculation of the systemic vascular resistance, how did the authors measure or estimate the central venous pressure?

- The paper would benefit from a more detailed description of the statistical models presented on page 17. In the equation, clearly define and reference all variables used within.

- The results section presents what appear to be the results of F-tests. This procedure should be described in the method’s section.

- The results section report the fluctuation of the hemodynamic and CV measures as measured with the “edf” values. The significance of this measurement was not clearly apparent from the discussion section, as it appeared that mainly steady-state values and transitions were of interest. What conclusions are drawn from the “edf” values? Also, hemodynamic and CV measures are known to fluctuate with the breathing, could this have played a role in the observed fluctuations?

- The statement “Previous research suggests that these lifestyle factors do not significantly alter the fundamental relationship between mechanisms governing CV processes and pain mechanisms.” requires citations.

- In the figures, the dashed vertical lines, which are used to delineate periods in that mark a significant between group differences, are hard to follows in some instances. The reviewer suggests modifying the graphical method used to highlight this difference.

- In the Supplemental Materials, the authors show figures comparing the Session 1 and Session 2 hemodynamic and CV measures. Certain measures, such as CO and SVR, show noticeable differences in overall magnitude, shapes, and effective degrees of freedom. Could the authors further comment on this.

Reviewer #2: I thank the authors for their very interesting contribution on a very practical topic for patients: studying the relationship between hemodynamic/cardiovascular mechanisms and chronic pain in search of CV-related 'pain-o-metric' indicators. The study utilizes patients with fibromyalgia as the diseased (in-pain) group and healthy individuals as control, recruiting an adequate number of patients for each (albeit homogeneous---which was nicely addressed in the limitations subsection). The entirely data-driven approach studies differences related to lying and standing (clinostatic and orthostatic challenges) between the groups in terms of SV, HR, CO, PEP, SVR, SBP, and DBP, highlighting between-group differences as a function of time during the challenges, and providing both basic/linear as well as covariate-adjusted results. A particular novelty here is in the use of non-linear statistical methods, which appears to provide a lot more insight than a standard linear data-driven approach (whose results the authors also provide for comparison). The results are convincing: there are clear differences between the FM and control groups, and the possible explanations in the discussion are credible in my opinion.

The paper is well-organized and excellently written (the English is nearly perfect, save for some typos I mention below), with a lot of comprehensive results and discussion. The latter, in particular, is relatively deep, with additional (alternative) arguments and discussions to be found in Supplementary Material. I also appreciate the unusually long "limitations" part of the discussion, which is very detailed.

I recommend this paper for publication so long as the following questions/comments/concerns are addressed. In particular, the methodology can get confusing and a bit vague, with not enough details/references.

(I'm sorry, but without line numbers, which I suggest you use next time, I can only give page numbers to direct you)

(these are in order of page number; the first few appear minor with copy edits, but eventually there are content-related questions/concerns in what follows)

1) Abstract: a couple of minor suggestions: "sec-by-sec" might be replaced by "second-by-second" (avoiding abbreviations in an abstract); "the chronic disease" might be replaced by "chronic disease" (no article); "enabled the identification" might be replaced by "enables..." (in order to be consistent with the present tense, and to respect that the methodology you are proposing is applicable beyond your work).

2) Intro, first paragraph, page 3: "World" should not be capitalized.

3) Table 1: there is no explanation, neither in the caption, nor in the text, of "t-\\chi" or "p-value" before the table appears. someone working in data-driven statistical approaches can guess easily, but for completeness, the former, at least, should be explained somewhere in text or caption.

4) Page 9: the title "Procedure, and clinical and psychological testing instruments" is a bit awkward; consider rephrasing to remove the second "and" (maybe: "Procedure and clinical/psychological testing instruments)

5) Page 12, bottom of the page: "respective data-driven adaptive techniques were used to take into account..." When someone reads this, they expect a list of the techniques, or more description. if you are referring to the rest of the "statistical metrics" section, then you should mention here ("as detailed in what follows")

6) Page 13, "hence, this non-linear methodology aimed to overcome..." It is not clear that your methodology is non-linear at this point. Also, the definition of "non-linear" versus "linear" is not mathematical here (I don't think), and so some phrases must be included in order to explain what exactly you mean by non-linear vs. linear. is non-linear just regressions that are co-variate adjusted?

7) Page 13, "because of this, a special non-parametric approximation technique..." What is the technique? Why is it special? Can you add some references? How do they accommodate non-linear effects?

8) Page 15, "fixed in 240" should be "fixed to 240"

9) On page 15, you italicize (correctly) "p" for p-value. elsewhere in the manuscript, it is no longer italisized (e.g. bottom of page 17). please correct throughout and make it consistent.

10) a number of metrics are used without citation...BIC, AIC. similarly, for bootstrapping discussion, please provide references and/or a description of what it exactly is for those who are less informed. in fact, a lot of the statistical methods section does not include references to the methods you are using. I know some are quite standard, but it would be useful for a reader to quickly refer to a paper for a description of the analysis metric you are using (i.e., to see the formulas, etc.)

11) "Supplementary materials" is sometimes lower case on the second word, sometimes upper case. Please correct throguhout.

12) It is unusual for someone to call Supplementary Materials Davydov (i.e., using their last name in the title). it's confusing. Please just use "Supplementary Materials" throughout the manuscript and in the supplement itself (i.e., remove your name from the section calls).

13) Page 17" "non-standardized (B) regression coefficients" ... shouldn't this "B" be lowercase to comply with convention?

14) in line with comment 9, perhaps consider italicizing (as is common) all the statistical variables (e.g., those that start on Page 18 results...b,t,p, etc.). it will make it easier to see these values embedded in such dense text.

15) Figures 1-7: since in a number of subfigures (particularly (B) and (C)) have teh exact same axes, a title might be useful to help people see right away that one plot is unadjusted and the other is adjusted. in fact, you may benefit from putting titles on all the figures (i.e., "Before adjustment of BMI", "After adjustment of BMI", etc.) In some cases it's (a) and (b), i.e., Figure 2. I think the specific co-variate considered (which varies between your figures/metrics) should be included in the figure titles.

16) Page 21: "with high, low, and very low HR frequency fluctuations" ...these are only two figures (2A) and (2B) corresponding to high and low fluctuations. please correct.

17) Figure 2, caption: I think there is a typo. in the last line, it should be "fast (C) and slow (D)" and not "fast (B) and slow (C)".

18) All figures: the grey dotted lines are not mentioned in any of the captions (as to what they demarcate). in the text they are not mentioned either, but we can easily infer that it's the intervals you are referring to. please metnion the dotted lines so people know what they mean more easily.

19) following 18), what do the dotted lines correspond to in Figure 2D? for other figures, you at least mention some of the intervals, or that you are showing locations of max deviation. but for 3D, I don't understand at all what the dotted lines are supposed to represent in the curves...

20) page 23, "very short random periods of significant differences": please revise. how do you define "significant difference" here, and "random periods"? i don't quite see that in the figure (the overall trend shows differences are high throughout...)

21) page 29, "after a common rapid SVR drops" should be "after a common rapid SVR drop" (singular)

22) figures 6 and 7: these appear to be the only two metrics (DBP and SBP) where, after co-variate adjustment, the differences betweem FM and CT become smaller. can you discuss/explain?

Reviewer #3: The manuscript presents an interesting and relevant investigation into the hemodynamic responses of fibromyalgia patients during clino-orthostatic challenges, which has important implications for understanding chronic pain and cardiovascular health. The use of a non-linear, data-driven technique seems a novel approach, and the study could potentially contribute valuable insights. However, I have some concerns that need to be addressed:

1- Many sentences specially in Introduction are overly long, which detracts from their clarity. I suggest revising these sentences to focus on one idea at a time. I suggest breaking up these sentences and focusing on presenting one idea at a time.

2- Please be consistent for the word “nonlinear” or “non-linear”. Choose one for the entire manuscript.

3- While the exclusion criteria are comprehensive, it would be helpful to provide more information on the specific FM-related medications allowed in the study. It would be important to clarify how the authors managed potential effects.How were their potential effects on cardiovascular outcomes accounted for?

4- The control group is described as being similar in age, nationality, civil status, and education level. However, it is not clear whether matching was done precisely or if this was achieved through random sampling. A more detailed explanation of the matching process would strengthen the methodological rigor of this section. For example, were controls individually matched to patients?

5- The exclusion criteria are well-detailed, but it would be beneficial to explain whether any additional steps were taken to control for confounding factors such as BMI, physical activity levels, or psychosocial stress, as these can influence cardiovascular and pain-related outcomes. Please update study limitations if needed.

6- A brief mention of how the sample size was determined or whether a power analysis was conducted would strengthen the validity of the study. Was the sample size based on previous studies in fibromyalgia, or calculated to ensure sufficient statistical power for the planned analyses?

7- It would be useful to specify if the tests were performed in the morning, afternoon, or at random times. Time of day can influence cardiovascular metrics, and standardizing test timing would reduce variability.

8- While participants were instructed to refrain from alcohol, smoking, caffeine, and exercise for three hours, three hours may be too short a time to eliminate the potential effects of these factors, especially caffeine. It would be beneficial to clarify whether this time period was based on previous studies or other justifications. Please update study limitations if needed.

9- Please mention whether potential confounders like sleep quality or daily pain fluctuations were considered when scheduling participants' appointments. Since FM patients often experience fluctuating symptoms, were steps taken to control for the time of day or participant-specific variability in pain levels? This could improve the consistency of physiological and psychological measurements.

10- The electrode placements for ECG and ICG are described in good detail, but providing a visual or diagram in the supplemental materials would help, particularly for those less familiar with these measurement techniques.

11- It would be useful to elaborate on the clinical implications of these findings. For example, how might these differences in SV response influence the management of FM patients in clinical settings?

12- The limitations of the study are well-addressed, but it could be useful to discuss any potential strategies for mitigating these limitations in future research.

13- Which cardiovascular metric you found most affected by pain? In total? In lying? And in Standing? Why do you think that metric is most affected?

14- Authors mentioned “Data supporting this study are not currently publicly available due to its farther use by the research team.”. At least, the data used for the plots in this study should be presented in Supplement I believe.

15- Could you explicitly indicate what is the novelty of this paper? Has any other study in the literature shown this novelty before? Please highlight the potential clinical significance of the key differences found in this study?

6. PLOS authors have the option to publish the peer review history of their article (what does this mean?). If published, this will include your full peer review and any attached files.

Reviewer #1: No

Reviewer #2: No

Reviewer #3: **Yes: **Rashid Alavi

---

## [Author Response · Author response to Decision Letter 0]

30 Oct 2024

Replies to comments

We sincerely appreciate the reviewers for their thoughtful comments and the editor for giving us the opportunity to resubmit the manuscript after revision. We have carefully addressed all the feedback and suggestions, making the necessary revisions to the manuscript. Below, we provide a point-by-point response to the editor's and reviewers' comments.

Editor’s comments:

Reply: Rechecked and corrected according to the PLOS ONE's style requirements.

Reply: No author-generated code was used in this study. All programs and procedures employed are publicly available, as detailed in the methods section.

3. Thank you for stating the following financial disclosure: “Gustavo A. Reyes del Paso, the Spanish Ministry of Science and Innovation, co-financed by European Regional Development Fund [PID2022-139731OB-I00]”. Please state what role the funders took in the study. If the funders had no role, please state: "The funders had no role in study design, data collection and analysis, decision to publish, or preparation of the manuscript." If this statement is not correct you must amend it as needed. Please include this amended Role of Funder statement in your cover letter; we will change the online submission form on your behalf.

Reply: The Role of the Funder statement has been included in both the cover letter and the corresponding section of the manuscript as follows: “The funders had no role in study design, data collection and analysis, decision to publish, or preparation of the manuscript.”

4. In this instance it seems there may be acceptable restrictions in place that prevent the public sharing of your minimal data. However, in line with our goal of ensuring long-term data availability to all interested researchers, PLOS’ Data Policy states that authors cannot be the sole named individuals responsible for ensuring data access (http://journals.plos.org/plosone/s/data-availability#loc-acceptable-data-sharing-methods). Data requests to a non-author institutional point of contact, such as a data access or ethics committee, helps guarantee long term stability and availability of data. Providing interested researchers with a durable point of contact ensures data will be accessible even if an author changes email addresses, institutions, or becomes unavailable to answer requests. Before we proceed with your manuscript, please also provide non-author contact information (phone/email/hyperlink) for a data access committee, ethics committee, or other institutional body to which data requests may be sent. If no institutional body is available to respond to requests for your minimal data, please consider if there any institutional representatives who did not collaborate in the study, and are not listed as authors on the manuscript, who would be able to hold the data and respond to external requests for data access? If so, please provide their contact information (i.e., email address). Please also provide details on how you will ensure persistent or long-term data storage and availability.

Reply: The requested contact information has been included in the manuscript under the data availability statement.

“Data requests may also be directed to the General Coordinator and Secretary of the Ethics Committee, referencing #: OCT.18/1.PRY and the title "Central Sensitization in Chronic Pain: The Case of Fibromyalgia": https://www.ujaen.es/gobierno/vicinv/comision-de-etica”

Replies to reviewers’ comments

Reviewer #1:

1 In the introduction, the authors claim that linear statistical methods were a limitation for analyzing the hemodynamic and CV reactions. As the methodological novelty of this paper originates from implementing non-linear models, it would be beneficial to expand on the current contributions of linear statistical models and where these have resulted in limited or erroneous understanding of hemodynamic processes.

Reply: Thank you for your comment. We have expanded this section accordingly (p. 5, lines 7-26, and p. 6, lines 1-7).

2 The statistical test performed and reported table 1 is not mentioned in the text. Please add a description of what has been performed and reported.

Reply: Thanks. Added.

3 The results of the administered questionnaires are presented in table 1, yet the description of these tests is described later in the manuscript. I suggest the authors consider reorganizing the flow.

Reply: The corresponding paragraphs were reorganized accordingly.

4 In the calculation of the systemic vascular resistance, how did the authors measure or estimate the central venous pressure?

Reply: We have addressed the issue on page 13, lines 5–7, with references supporting the point.

5 The paper would benefit from a more detailed description of the statistical models presented on page 17. In the equation, clearly define and reference all variables used within.

Reply: This point is now addressed in more details on pages 19-20, with references to other paragraphs of the section (pages 15 and 16) where these models and the variables included in the equation are discussed and further clarified.

6 The results section presents what appear to be the results of F-tests. This procedure should be described in the method’s section.

Reply: This point is now addressed on page 19 (lines 11-12).

7 The results section report the fluctuation of the hemodynamic and CV measures as measured with the “edf” values. The significance of this measurement was not clearly apparent from the discussion section, as it appeared that mainly steady-state values and transitions were of interest. What conclusions are drawn from the “edf” values? Also, hemodynamic and CV measures are known to fluctuate with the breathing, could this have played a role in the observed fluctuations?

Reply: Thank you for your interest in the indicator. Due to the manuscript's length, we initially limited our discussion of this point to a brief remark (see page 52, lines 1-21 of the revised version). However, we have now expanded the discussion to emphasize the additional value of this measure, providing its short analysis at the end of the subsections discussing findings related to each hemodynamic and cardiovascular variable, with further details now included in S Materials.VIII.

8 The statement “Previous research suggests that these lifestyle factors do not significantly alter the fundamental relationship between mechanisms governing CV processes and pain mechanisms.” requires citations.

Reply: Thank you. In the cited version of the manuscript, these references were originally placed after the following sentence, which further elaborates on the issue. We have now moved them to the indicated sentence (page 54, line 6).

9 In the figures, the dashed vertical lines, which are used to delineate periods in that mark a significant between group differences, are hard to follows in some instances. The reviewer suggests modifying the graphical method used to highlight this difference.

Reply: Thank you for your comment. The original high-resolution figures included additional markers indicating areas of significant differences, which were unfortunately not visible in the low-resolution versions. We apologize for this oversight. We have now uploaded the high-resolution figures, where the time windows showing significant between-group differences (FM vs. CT) are marked with horizontal bold-line projections on the Time (X) axis. These are bounded by vertical dashed lines, corresponding to the relevant points along the CV difference curves. The figure legends have also been updated to reflect this clarifying information.

10 In the Supplemental Materials, the authors show figures comparing the Session 1 and Session 2 hemodynamic and CV measures. Certain measures, such as CO and SVR, show noticeable differences in overall magnitude, shapes, and effective degrees of freedom. Could the authors further comment on this.

Reply: Thank you for your interest in the findings. The section S Materials.I has been enhanced with additional discussions on within-subject effects on hemodynamics across sessions (pages 3-5).

Reviewer #2:

1. Abstract: a couple of minor suggestions: "sec-by-sec" might be replaced by "second-by-second" (avoiding abbreviations in an abstract); "the chronic disease" might be replaced by "chronic disease" (no article); "enabled the identification" might be replaced by "enables..." (in order to be consistent with the present tense, and to respect that the methodology you are proposing is applicable beyond your work).

Reply: Thank you! The references in the abstract have been corrected.

2. Intro, first paragraph, page 3: "World" should not be capitalized.

Reply: The word has been corrected.

3. Table 1: there is no explanation, neither in the caption, nor in the text, of "t-\\chi" or "p-value" before the table appears. Someone working in data-driven statistical approaches can guess easily, but for completeness, the former, at least, should be explained somewhere in text or caption.

Reply: Thank you! The necessary explanation has been included in the caption and added to the methods section (page 20, lines 13-16).

4. Page 9: the title "Procedure, and clinical and psychological testing instruments" is a bit awkward; consider rephrasing to remove the second "and" (maybe: "Procedure and clinical/psychological testing instruments)

Reply: Thank you for the suggestion! The title has been corrected.

5. Page 12, bottom of the page: "respective data-driven adaptive techniques were used to take into account..." When someone reads this, they expect a list of the techniques, or more description. if you are referring to the rest of the "statistical metrics" section, then you should mention here ("as detailed in what follows")

Reply: Thank you for the suggestion! The sentence has been corrected.

6. Page 13, "hence, this non-linear methodology aimed to overcome..." It is not clear that your methodology is non-linear at this point. Also, the definition of "non-linear" versus "linear" is not mathematical here (I don't think), and so some phrases must be included in order to explain what exactly you mean by non-linear vs. linear. is non-linear just regressions that are co-variate adjusted?

Reply: Thank you for highlighting this issue. The indicated sentence, along with the entire paragraph, was indeed misplaced and has now been relocated after the following paragraph, combining them into one (page 14, lines 21-25, and page 15, lines 1-17) to improve the clarity of the discourse.

7. Page 13, "because of this, a special non-parametric approximation technique..." What is the technique? Why is it special? Can you add some references? How do they accommodate non-linear effects?

Reply: We have included (page 14, lines 20-22) the reference to the package implementing the technique used, along with a section below that elaborates on the details of the technique.

8. Page 15, "fixed in 240" should be "fixed to 240"

Reply: Thanks. Corrected.

9. On page 15, you italicize (correctly) "p" for p-value. elsewhere in the manuscript, it is no longer italisized (e.g. bottom of page 17). please correct throughout and make it consistent.

Reply: All relevant designations have been italicized throughout the manuscript.

10. a number of metrics are used without citation...BIC, AIC. similarly, for bootstrapping discussion, please provide references and/or a description of what it exactly is for those who are less informed. in fact, a lot of the statistical methods section does not include references to the methods you are using. I know some are quite standard, but it would be useful for a reader to quickly refer to a paper for a description of the analysis metric you are using (i.e., to see the formulas, etc.)

Reply: We added more references to literature about the primary mathematical and statistical methods to read along with references to secondary methods.

11. "Supplementary materials" is sometimes lower case on the second word, sometimes upper case. Please correct throughout.

Reply: Thank you! Corrected.

12. It is unusual for someone to call Supplementary Materials Davydov (i.e., using their last name in the title). it's confusing. Please just use "Supplementary Materials" throughout the manuscript and in the supplement itself (i.e., remove your name from the section calls).

Reply: Corrected.

13. Page 17" "non-standardized (B) regression coefficients" ... shouldn't this "B" be lowercase to comply with convention?

Reply: Corrected.

14. in line with comment 9, perhaps consider italicizing (as is common) all the statistical variables (e.g., those that start on Page 18 results...b,t,p, etc.). it will make it easier to see these values embedded in such dense text.

Reply: Corrected.

15. Figures 1-7: since in a number of subfigures (particularly (B) and (C)) have the exact same axes, a title might be useful to help people see right away that one plot is unadjusted and the other is adjusted. in fact, you may benefit from putting titles on all the figures (i.e., "Before adjustment of BMI", "After adjustment of BMI", etc.) In some cases it's (a) and (b), i.e., Figure 2. I think the specific co-variate considered (which varies between your figures/metrics) should be included in the figure titles.

Reply: Added to the respective figures to indicate their relevance to analyses before or after adjustments, if permitted by editorial policy, but without specifying particular covariates, as this information is already provided in the figure captions.

16. Page 21: "with high, low, and very low HR frequency fluctuations" ...these are only two figures (2A) and (2B) corresponding to high and low fluctuations. please correct.

Reply: Thank you! Corrected.

17. Figure 2, caption: I think there is a typo. in the last line, it should be "fast (C) and slow (D)" and not "fast (B) and slow (C)".

Reply: Thank you! Yes, you are right! Corrected.

18. All figures: the grey dotted lines are not mentioned in any of the captions (as to what they demarcate). in the text they are not mentioned either, but we can easily infer that it's the intervals you are referring to. please mention the dotted lines so people know what they mean more easily.

Reply: Thank you! The clarification for the vertical dashed lines has been added to the corresponding figures.

19. following 18), what do the dotted lines correspond to in Figure 2D? for other figures, you at least mention some of the intervals, or that you are showing locations of max deviation. but for 3D, I don't understand at all what the dotted lines are supposed to represent in the curves...

Reply: We have added further clarifications for the vertical dashed lines and the regions they indicate, along with the horizontal bold lines on the X-axis that were not visible in the low-resolution figures. We apologize for the inconsistency.

20. page 23, "very short random periods of significant differences": please revise. how do you define "significant difference" here, and "random periods"? i don't quite see that in the figure (the overall trend shows differences are high throughout...)

Reply: The sentence has been revised (page 25, lines 8-10).

21. page 29, "after a common rapid SVR drops" should be "after a common rapid SVR drop" (singular)

Reply: Thank you! Corrected (page 31, line 10).

22. figures 6 and 7: these appear to be the only two metrics (DBP and SBP) where, after co-variate adjustment, the differences between FM and CT become smaller. can you discuss/explain?

---

## [Decision Letter · Decision Letter 1]

25 Nov 2024

Hemodynamics in chronic pain: A pathway to multi-modal health risks

PONE-D-24-27267R1

Dear Dr. Davydov,

We’re pleased to inform you that your manuscript has been judged scientifically suitable for publication and will be formally accepted for publication once it meets all outstanding technical requirements.

Kind regards,

Niema M. Pahlevan, PhD

Academic Editor

PLOS ONE

Additional Editor Comments (optional):

Reviewers' comments:

Reviewer's Responses to Questions

**Comments to the Author**

1. If the authors have adequately addressed your comments raised in a previous round of review and you feel that this manuscript is now acceptable for publication, you may indicate that here to bypass the “Comments to the Author” section, enter your conflict of interest statement in the “Confidential to Editor” section, and submit your "Accept" recommendation.

Reviewer #1: All comments have been addressed

Reviewer #3: All comments have been addressed

2. Is the manuscript technically sound, and do the data support the conclusions?

Reviewer #1: Yes

Reviewer #3: Yes

3. Has the statistical analysis been performed appropriately and rigorously? 

Reviewer #1: Yes

Reviewer #3: Yes

4. Have the authors made all data underlying the findings in their manuscript fully available?

Reviewer #1: Yes

Reviewer #3: No

5. Is the manuscript presented in an intelligible fashion and written in standard English?

Reviewer #1: Yes

Reviewer #3: Yes

6. Review Comments to the Author

Reviewer #1: The authors has adequately addressed all of the comments from the previous round of revision. This reviewer has no further comments.

Reviewer #3: All my comments have been addressed, no more comments. The manuscript looks good to me now.Thank you!

7. PLOS authors have the option to publish the peer review history of their article (what does this mean?). If published, this will include your full peer review and any attached files.

Reviewer #1: No

Reviewer #3: **Yes: **Rashid Alavi

---

## [Editor Report · Acceptance letter]

29 Nov 2024

PONE-D-24-27267R1 

PLOS ONE

Dear Dr. Davydov, 

I'm pleased to inform you that your manuscript has been deemed suitable for publication in PLOS ONE. Congratulations! Your manuscript is now being handed over to our production team.

Kind regards, 

on behalf of

Dr. Niema M. Pahlevan 

Academic Editor

PLOS ONE